# Simultaneous profiling of RNA isoforms and chromatin accessibility of single cells of human retinal organoids

Shuyao Zhang [1,5], Yuhua Xiao [1,5], Xinzhi Mo [1,5], Xu Chen [1], Jiawei Zhong [1], Zheyao Chen [1], Xu Liu [1], Yuanhui Qiu [1], Wangxuan Dai [1], Jia Chen [1], Xishan Jin [1], Guoping Fan [2,3] & Youjin Hu [1,4] ✉

Single-cell multi-omics sequencing is a powerful approach to analyze complex mechanisms underlying neuronal development and regeneration. However, current methods lack the ability to simultaneously profile RNA alternative splicing and chromatin accessibility at the single-cell level. We develop a technique, single-cell RNA isoform and chromatin accessibility sequencing (scRICA-seq), which demonstrates higher sensitivity and cost-effectiveness compared to existing methods. scRICA-seq can profile both isoforms and chromatin accessibility for up to 10,000 single cells in a single run. Applying this method to human retinal organoids, we construct a multi-omic cell atlas and reveal associations between chromatin accessibility, isoform expression of fate-determining factors, and alternative splicing events in their binding sites. This study provides insights into integrating epigenetics, transcription, and RNA splicing to elucidate the mechanisms underlying retinal neuronal development and fate determination.

RNA isoform diversity mediates the variable functions of individual genes, accounting for the complex phenotypes of cells[1,2]. Epigenetic modifications also significantly contribute to the heterogeneity among single cells, which has been recently found to mediate alternative splicing (AS) events. However, efficient methods to profile RNA isoforms and epigenetic status of individual single cells genome-wide are still lacking. Although many single-cell sequencing methods to profile transcriptomes and epigenomes have been developed[3–6], a method to simultaneously detect RNA isoforms and epigenomes has not yet been developed.

The majority of current single-cell RNA sequencing methods are based on short-read sequencing, with read lengths below 150 base pairs, which is not suitable for characterizing the structures of full-length RNA. Long-read sequencing-based single-cell RNA sequencing methods can reveal full-length RNA isoforms, but the high cost, lower sequencing depth, and reduced sensitivity make them less applicable for integration into single-cell multi-omics analysis[7–9].

Previously, we develop a short-read-based method, scRCAT-seq, to profile the diversities in transcription start sites (TSS) and transcription end sites (TES) of RNA isoforms, which requires only nanograms of cDNA as input[10]. Here, we optimize the protocol of scRCAT-seq to profile the full-length of single molecules of RNA at a genome-wide scale (scRCAT-seq2). Furthermore, we integrate this approach with single-cell ATAC-seq to simultaneously profile chromatin accessibility and full-length RNA isoforms within the same single cells (scRICA-seq, single-cell RNA isoform and chromatin accessibility sequencing). By applying scRICA-seq to profile the multi-omics landscape of human retinal organoids, we reveal concordant regulation of chromatin accessibility and RNA splicing on key fate-determinant genes mediating neuronal development and regeneration within the human retinal organoids.

[1]State Key Laboratory of Ophthalmology, Zhongshan Ophthalmic Center, Sun Yat-Sen University, Guangzhou, China. [2]Department of Human Genetics, David Geffen School of Medicine, UCLA, Los Angeles, CA, USA. [3]Scintillon Research Institute, 6868 Nancy Ridge Drive, San Diego, CA 92121, USA. [4]Guangdong Provincial Key Laboratory of Ophthalmology and Visual Science, Guangzhou, China. [5]These authors contributed equally: Shuyao Zhang, Yuhua Xiao, Xinzhi Mo. ✉e-mail: huyoujin@gzzoc.com

## Results

### Rationale of scRICA-seq

Based on the parallel capture of DNA and RNA from the same single cells, scRICA-seq is designed to simultaneously profile the chromatin accessibility, RNA expression, and isoform expression within the same cell. We employ two strategies that enable the parallel capture and sequencing of DNA and RNA. For low-throughput single-cell samples, we utilize a nuclear-cytoplasmic separation method that has been previously established by our team[11]. This method allows us to obtain cell nuclei and cytoplasmic RNA from the same cell (Supplementary Fig. 1a). For high-throughput single-cell populations, we employ a microfluidic system, which enables the parallel analysis of chromatin accessibility and transcriptome within individual cell nuclei (Fig. 1a, Supplementary Fig. 2).

To profile the single-cell transcriptome of full-length isoforms, we developed a method named scRCAT-seq2, which builds upon our previous work on scRCAT-seq[10]. The scRCAT-seq2 method encompasses the following steps: (1) Labeling the ends of individual RNA/cDNA molecules with a unique molecular identifier (UMI); (2) Creating a circular cDNA through end-to-end ligation; (3) Performing circular amplification to generate multiple copies of full-length cDNA; (4) Fragmenting the full-length cDNA by randomly inserting Tn5 transposases across the circular full-length cDNA; (5) Constructing libraries and conducting paired-end sequencing on the resulting fragments; (6) Integrating short reads with the same UMI, which were collected and mapped to specific features of the annotated individual isoforms, including the transcription start sites (TSS), transcription end sites (TES), and specific exons. Each UMI was then assigned to an individual isoform with specific features covered by the sequencing reads (Fig. 1b, Supplementary Figs. 1b, 2, 3).

### Performance of scRCAT-seq2

The results show that the short reads of scRCAT-seq2 can effectively cover the full-length transcripts, including the TSS, TES, and exons (Fig. 2a, Supplementary Fig. 4). To evaluate the sensitivity and accuracy of scRCAT-seq2 in identifying and quantifying isoforms within single cells, we spiked in the RNA standard set (SIRV-set 3) containing 92 ERCCs and 69 isoforms to serve as a ground-truth control. ScRCAT-seq2 can identify 61 SIRV transcripts, with a sensitivity of 88%, and the full-length features were revealed with an accuracy of 100% (Fig. 2b, c). Furthermore, we observed a high correlation of ERCC expression between individual single cells, and high consistency of the observed abundance with the expected values (Fig. 2d, e). We also found a strong correlation of genes and isoforms between individual single cells (Fig. 2f, Supplementary Figs. 5, 6a), indicating high consistency of the scRCAT-seq2 data.

Further, we compared the performance of scRCAT-seq2 to other methods capable of detecting isoforms at the single-cell level, including PacBio long-read sequencing-based ScISOr-seq[8] and HIT-scISO-seq[12], Nanopore long-read sequencing-based SCAN-seq2[13] and scCOLOR-seq[9], and short-read sequencing-based scRCAT-seq[10]. We observed the highest sensitivity and efficiency in detecting single-cell isoforms by scRCAT-seq2 across different sequencing depths of equal cost (Fig. 2g). Specifically, when the single-cell cDNA is generated in low-throughput, the saturated sequencing depth of scRCAT-seq2 approaches, and it can detect 12,695 ± 348 isoforms in ES cells, which is significantly more compared to other approaches, including ScISOr-seq (1041 ± 41), SCAN-seq2 (8566 ± 123), and scRCAT-seq (2152 ± 130) (Fig. 2h, Supplementary Fig. 6b, d). For the high-throughput single-cell samples, where the cDNA was amplified on microfluidic platforms, scRCAT-seq2 also shows a higher number of isoforms compared to HIT-scISO-seq and Nanopore-based methods in monkey corneal limbal epithelium cells and human retinal organoids, respectively (Fig. 2i, Supplementary Fig. 6c). Finally, we tested scRCAT-seq on frozen samples of mouse kidney. The results show high-quality cDNA and recovered thousands of genes and isoforms. The similarity between the transcriptomic data of different single cells is comparable to fresh samples (Spearman correlation coefficient >0.8), indicating that this method can be applicable to frozen samples (Supplementary Fig. 7).

In summary, these results demonstrated the high sensitivity, consistency and efficiency of scRCAT-seq2 to profile the single cell isoforms.

### A multi-omics cell atlas of human retinal organoids

By integrating the scRCAT-seq2 and scATAC-seq, scRICA-seq can analyze the heterogeneity and correlation among isoform expression and chromatin accessibility at the single-cell level (Supplementary Fig. 8). To showcase its application, we applied scRICA-seq to human retinal development and constructed a single-cell multi-omics atlas of human retinal organoids (Fig. 1a, Supplementary Fig. 8). The clustering results based on single-cell isoforms identified 6 clusters, and successfully detected various cell types with typical cell markers, including retinal progenitor cells (RPCs), retinal ganglion cells (RGCs), amacrine cells/horizontal cells (ACs/HCs), photoreceptor precursors (PR precursors), neurogenic RPCs, and cones, which is highly consistent with those obtained from scRNA-seq (Fig. 3a–c). In addition, we identified cell type specific features of gene and isoform expression, and chromatin accessibility levels at the corresponding promoters, such as the *THRB*[10,14] in photoreceptor precursor cells, *CRX*[15,16], *PDE6H*[17], and *ARR3*[18,19] in cones, *PAX6*[20,21] in ACs/HCs, and *GAP43*[22], *ISL1*[23] in RGCs (Fig. 3d–g, Supplementary Figs. 9, 10). Furthermore, we also observed fate-determining factors of retinal neurons, such as *HES1* in RPCs[24], *AHOH7* in neurogenic RPCs[24–26], *ISL1*[23] in RGCs, *CRX* in the PR precursor (Fig. 3d–g), and the RNA expression and chromatin accessibility patterns of these genes were aligned with their activity in different cell types (Fig. 3g). This suggests that chromatin accessibility plays a role in mediating the expression of marker genes and fate-determination factors.

To further reveal the correlation between isoform expression and chromatin accessibility during differentiation, we performed pseudo-time differentiation trajectory analysis for RPCs, neurogenic RPCs, PR precursors, and cones (Fig. 4a, b, Supplementary Fig. 11), and observed many genes with expression and promoter accessibility consistently changing during the differentiation of RPCs into cones. These include the regulatory transcription factor *HES1*, *OTX2*, *CRX*, *NRL*, *ATOH7*, and *THRB*, showing concordant changes in gene and isoform expression, as well as chromatin accessibility of the promoter during differentiation. (Fig. 4c–f). For fate-determining factors, including *NRL*, *ARR3*, *CRX*, and *THRB*, we observed the chromatin accessibility of promoter increased prior to the expression changes of the corresponding isoforms, suggesting that the changed chromatin accessibility set the promoters into a poised state before transcriptional activation (Fig. 4g–i, Supplementary Fig. 12). Previous studies have confirmed that the expression of *NRL* is regulated by H3K27 epigenetic modifications[27], and the function of H3K27 in regulating chromatin accessibility and poised state has been demonstrated in various tissues including the nervous system and tumors[28,29]. We hypothesize that the open chromatin accessibility is a necessary factor for the activation of certain isoforms of these fate-determination factors (Fig. 4g–i, Supplementary Fig. 12).

In addition, we observed that chromatin accessibility in the binding regions of some key fate-determining factors was generally positively correlated with the expression levels of their corresponding genes. Notably, the significance of this correlation varies, suggesting differential contribution of chromatin accessibility to the transcriptional regulation among these transcription factors (Fig. 4j–l, Supplementary Figs. 13, 14).

In summary, by using scRICA-seq, we constructed a multi-omics cell atlas of human retinal organoids, and revealed the correlations between chromatin accessibility, gene expression, and isoform expression during retinal neuron development.

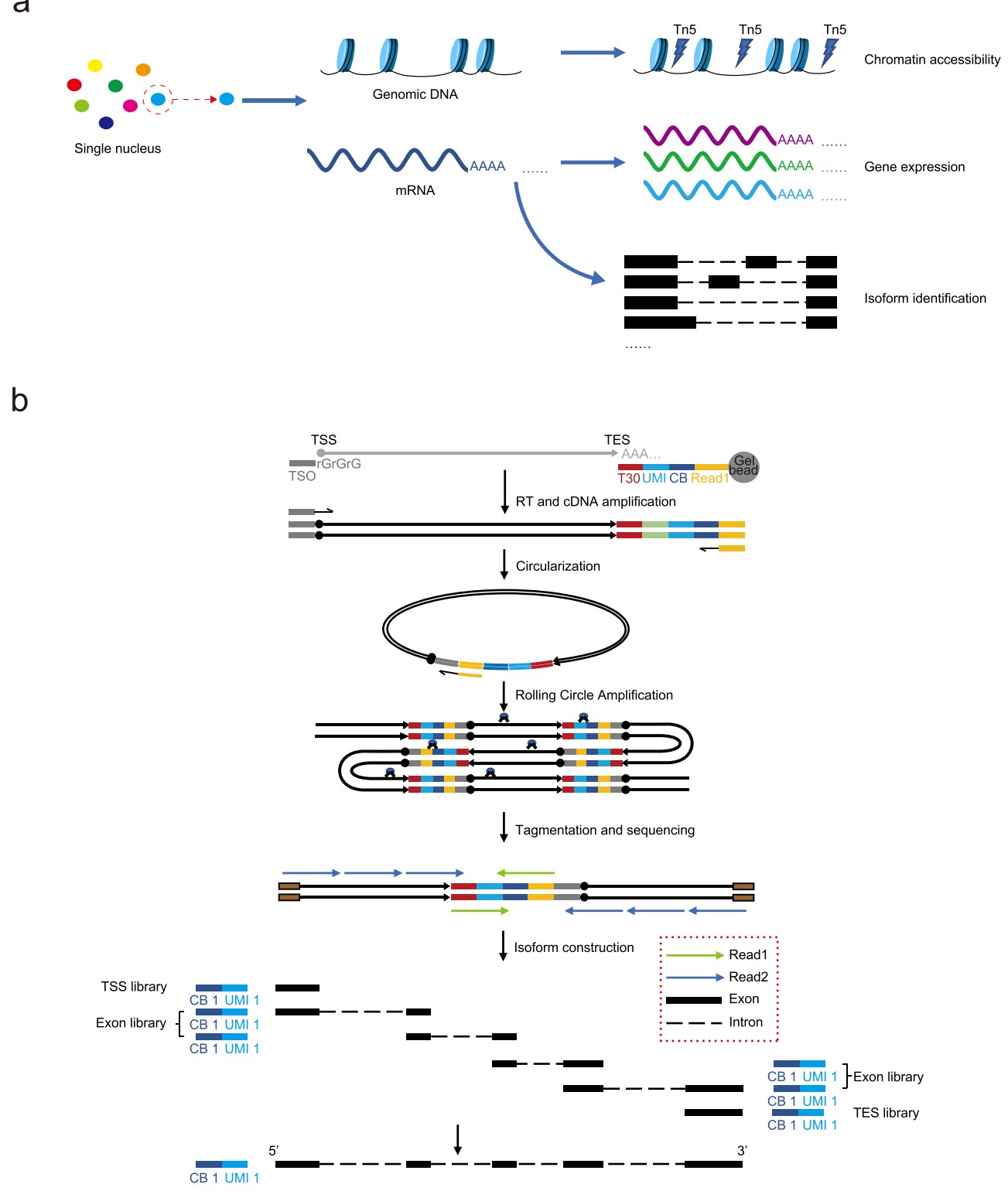

**Fig. 1 | Overview of scRICA-seq in high throughput. a** Schematic illustration of scRICA-seq to simultaneously profile the RNA isoforms and chromatin accessibility for the same single nucleus, by integrating the scRNA-seq, scATAC-seq, scRCAT- seq2. **b** Schematic illustration of scRCAT-seq2 to depict the step-by-step procedure for generating the full-length cDNA library. CB: Cell barcode; UMI: Unique molecular identifier; TSO: Template-switching oligo; T30: 30 repeating T bases.

## Interplay between chromatin accessibility and alternative splicing during retinal progenitor cell differentiation

We examined the dynamic patterns of RNA isoform usage during the differentiation of RPCs into cone photoreceptors. This comprehensive analysis encompassed various types of alternative splicing events, including alternative promoters, alternative polyadenylation, exon skipping (SE), intron retention (RI), mutually exclusive exons (MXE), and alternative 3′ / 5′ splice site selection (A3SS/A5SS)[30]. Our results identified a total of 2292 differentially expressed isoforms between mature cones and RPCs, with 1143 isoforms highly

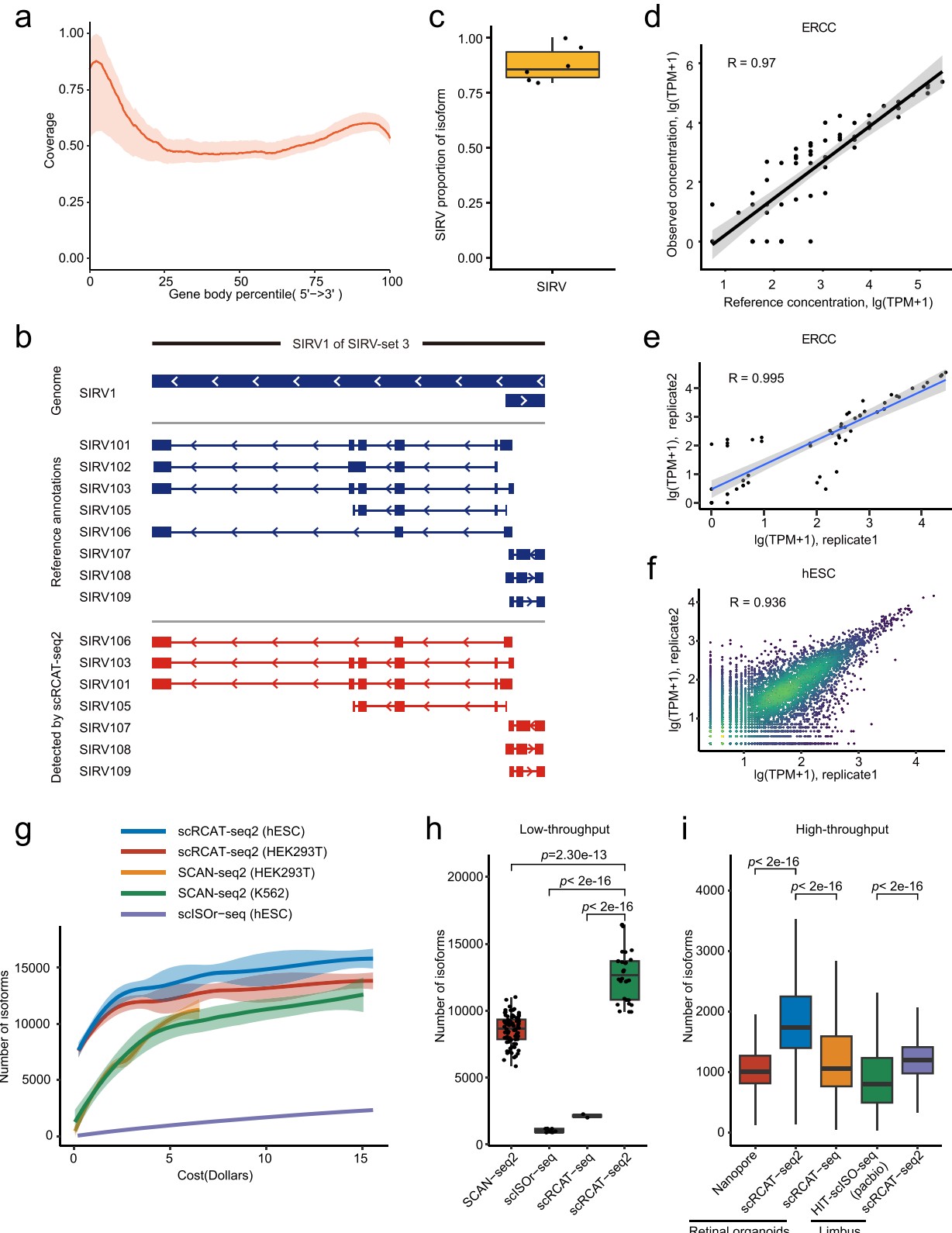

expressed in cones and 1149 in RPCs (Fig. 5a, Supplementary Fig. 15a, b).

Furthermore, our analysis revealed that 7.43% of the splicing sites exhibited significant changes in chromatin accessibility during RPC differentiation into cone photoreceptors (Fig. 5b). By performing transcription factor motif analysis, we predicted 589 transcription factors that may bind to these dynamically regulated splicing sites.

Specifically, the regions with increased chromatin accessibility near the dynamic splicing sites were enriched with 78 transcription factors, while the regions with decreased chromatin accessibility were enriched with 57 transcription factors (Supplementary Fig. 16). Among the predicted transcription factors, we identified 11 that have been previously reported to be directly involved in regulating RNA splicing, including *NEUROD1*[31,32], *SP1*, and *KLF4*[33] (Fig. 5c). Interestingly, we also

**Fig. 2 | Performance of scRCAT-seq2. a** Coverage of scRCAT-seq2 data on transcripts of single ES cells (*n* = 6, A total of 6 replicates of scRCAT-seq2 were performed, each time using one ES cell with SIRV-Set 3). The displayed values represent the averaged coverage of reads with shaded areas indicating standard deviations. **b** In SIRV-set 3, the gene SIRV1 is demonstrated as an example, where the full-length transcriptome was identified and constructed through scRCAT-seq2. The center line: median; boxes: first and third quartiles; whiskers: 5th and 95th percentiles. **c** Sensitivity of scRCAT-seq2 in identifying transcripts in SIRV-Set 3, with 88% (61 isoforms were characterized from 69 isoforms) of the transcripts detected. Error bars represent the standard deviation of the mean (*n* = 6, A total of 6 replicates of scRCAT-seq2 were performed, each time using one ES cell with SIRV-Set 3). **d** Correlation scatter plot between the expected (*x*-axis) and observed abundance (*y*-axis) of ERCC transcripts (log-transformed, Pearson correlation coefficient *r* = 0.97, *n* = 92, A total of 92 ERCC transcripts). **e** Correlation scatter plot of ERCC transcripts between replicates (Pearson correlation coefficient *r* = 0.995, *n* = 6). **f** Correlation scatter plot of gene expression levels in hESC samples between replicates (Pearson correlation coefficient *r* = 0.936, *n* = 6). **g** The number of isoforms detected using scRCAT-seq2 (hESC, *n* = 6; HEK293T, *n* = 6), SCAN-seq (HEK293T, *n* = 6; K562, *n* = 6), or scISOr-seq (hESC, *n* = 2), versus cost. Displayed is the mean number of transcripts with 95% confidence intervals shaded. **h** Comparison of the number of transcripts detected by scRCAT-seq2 (*n* = 36, A total of 36 ES cells and 293T cells with the same sequencing cost), SCAN-seq (*n* = 240, A total of 240 K562 cells and 293T cells with the same sequencing cost), scISOr-seq (*n* = 2, A total of 2 ES cells with the same sequencing cost), and scRCAT-seq (*n* = 2, A total of 2 ES cells with the same sequencing cost) at the same cost in low-throughput sequencing. Significance was computed using two-sided *t*-test. The center line: median; boxes: first and third quartiles; whiskers: 5th and 95th percentiles. **i** Comparison of the number of isoforms detected by scRCAT-seq2, nanopore, and scRCAT-seq in high-throughput sequencing of retinal organoids; comparison of the number of isoforms detected by scRCAT-seq2 and HIT-scISO-seq in high-throughput sequencing of monkey corneal limbal epithelium cells (Limbus). Significance was computed using two-sided *t*-test. The center line: median; boxes: first and third quartiles; whiskers: 5th and 95th percentiles. Source data are provided as a Source Data file.

found 10 transcription factors that have been previously shown to regulate cell fate-determining factors, such as *NEUROD1*, *ASCL1*, and *THRB* (Fig. 5d). This suggests that dynamic changes in chromatin accessibility at splicing sites may allow these transcription factors to coordinately control both splicing programs and cell fate decisions during retinal progenitor cell differentiation.

Further investigation revealed that some of the splicing-related transcription factors, such as *NEUROD1*, exhibited co-enrichment at the chromatin accessibility-differential sites, implying cooperative regulation of differential RNA splicing (Fig. 5e). For example, the transcription factor *NEUROD1*, which has been previously reported to be associated with RNA splicing[32], is related to the differential isoform choice of many genes in the retinal organoids, including *MTHFD2*, *MAP3K3*, *MXRA7*, and *CRX* (Fig. 5f–h, Supplementary Fig. 15c–f). Together, these findings highlight the intricate interplay between chromatin accessibility dynamics and alternative splicing programs that drive retinal progenitor cell differentiation into cone photoreceptors.

Importantly, we also observed differential isoform selection in regions where chromatin accessibility did not change, indicating that additional mechanisms beyond chromatin accessibility dynamics may also regulate alternative splicing. For example, we detected differential expression of known splicing factors, such as *KDM5B* and *CELF1*, between RPCs and cone photoreceptors. Furthermore, the differential expression of transcription factors, including *NEUROD1* and *ATF2*, was associated with RNA splicing at their binding sites (Supplementary Fig. 15f, g). Interestingly, we found that the expression of these transcription factors was significantly correlated with the chromatin accessibility at their promoters (Supplementary Fig. 17). This suggests that epigenetic regulation of these splicing-related transcription factors may also contribute to the control of alternative splicing programs during retinal cell differentiation.

In summary, scRCAT-seq2 provides an unprecedented opportunity to integrate the analysis of chromatin accessibility with the dynamic changes in isoforms at single cell level. The splicing isoforms and related fate-determining factors identified in this study provide clues for integrating epigenetics, RNA transcription, and splicing research to elucidate the mechanisms underlying retinal neuronal development and fate determination.

## Discussion

This study introduces a single-cell multi-omics method to enable a comprehensive analysis on the correlation between chromatin accessibility, RNA expression, and differential isoform choices. In addition, we present a framework to investigate the potential mechanisms underlying retinal neuron development and regeneration by integrating chromatin accessibility, gene expression and splicing features.

Majority of existing scRNA-seq methods capture short fragments from either the 5′ end or 3′ end of transcripts with limited ability to profile the full-length features. Although imputation methods, like Scasa[34], have been developed to predict isoform expression with short-read sequencing data, the accuracy and efficiency are still largely limited due to lack of important features away from the 3′ end[34]. While a few methods can capture and amplify the full-length cDNA of single cells, including smart-seq[35], smart-seq2[36], smart-seq3[37], PMA and SMA[38], but due to the incapability to link internal short reads to the UMI located in the 5′/3′ end, the efficiency to identify full-length isoforms is still largely limited.

Furthermore, scRCAT-seq2 outperforms the long-read sequencing-based methods in terms of cost-effectiveness. Our data shows that the saturation sequencing depth for scRCAT-seq2 is 4 million reads (equal to 4 dollars), while for nanopore based SCAN-seq is 0.4 million (equal to 7 dollars), and for the Pacbio sequencing based scISOr-seq is 0.4 million (equal to 120 dollars). This indicates that scRCAT-seq2 has a significantly lower saturation sequencing depth and much higher read coverage. Compared with the emerging method based on the third-generation sequencing technology, scRCAT-seq2 has higher sensitivity than ScISOr-seq and other technologies can realize the analysis of full-length transcripts in single cells such as a large number of cells, but the average number of transcripts per single cell is only a few hundred[8,39], which is far below the level of 12,000 transcripts that can be detected by this method. Recently, SCAN-seq[40] and SCAN-seq2[13] can increase the number of isoforms detected in a single cell, but they require library construction in a single cell unit, which is labor-intensive and difficult to increase the throughput (Supplementary Data 1).

Importantly, our method extends the analysis of differential isoform expression beyond genetic mutations to epigenetic changes by profiling the RNA isoforms and chromatin accessibility with the same single cells. This method not only reveal the precise linkage of the chromatin accessibility status with expression level of individual isoforms, but also gain insights into the alternative splicing correlated with epigenetic state. we observe changes in chromatin accessibility states near multiple variable splicing sites, not only validating the hypothesis that chromatin accessibility can regulate splicing by modulating transcription rates, but also identifying the binding of transcription factors such as *NEUROD1*, *KLF4*, *SP1*, and *ZNF263* near these variable splicing sites. This leads us to propose that the binding of these transcription factors may be involved in the regulation of transcriptional splicing, providing insights into the mechanisms underlying retinal neuronal development and fate determination by integrating epigenetics, RNA transcription, and splicing information.

Our study also has several limitations. First, despite the unprecedented sensitivity to profile the full-length transcripts, scRCAT-seq2

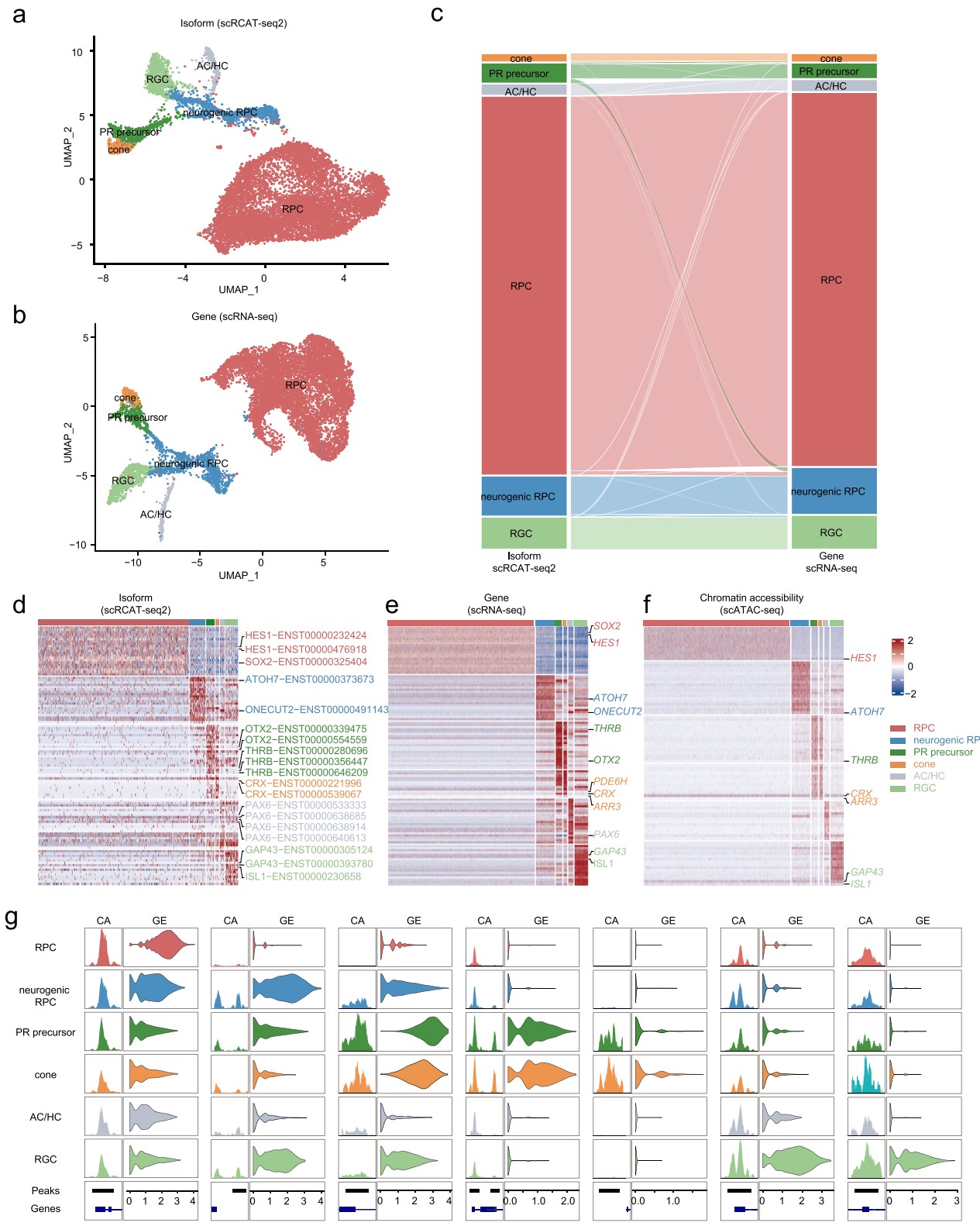

**Fig. 3 | A multi-omics single-cell atlas of human retinal organoids based on scRICA-seq. a** Cell clustering of human retinal organoids with the scRCAT-seq2 data. **b** Cell clustering of human retinal organoid by using scRNA-seq. **c** Matching the clusters identified by scRCAT-seq2 and scRNA-seq respectively. **d** Heatmap displaying marker isoforms of each cluster detected by scRCAT-seq2, (**e, f**) Heatmap displaying marker genes of each cluster detected by scATAC-seq and scRNA-seq respectively. **g** Visualization of the chromatin accessibility (CA) and gene expression (GE) of representative marker gene loci in annotated cell clusters.

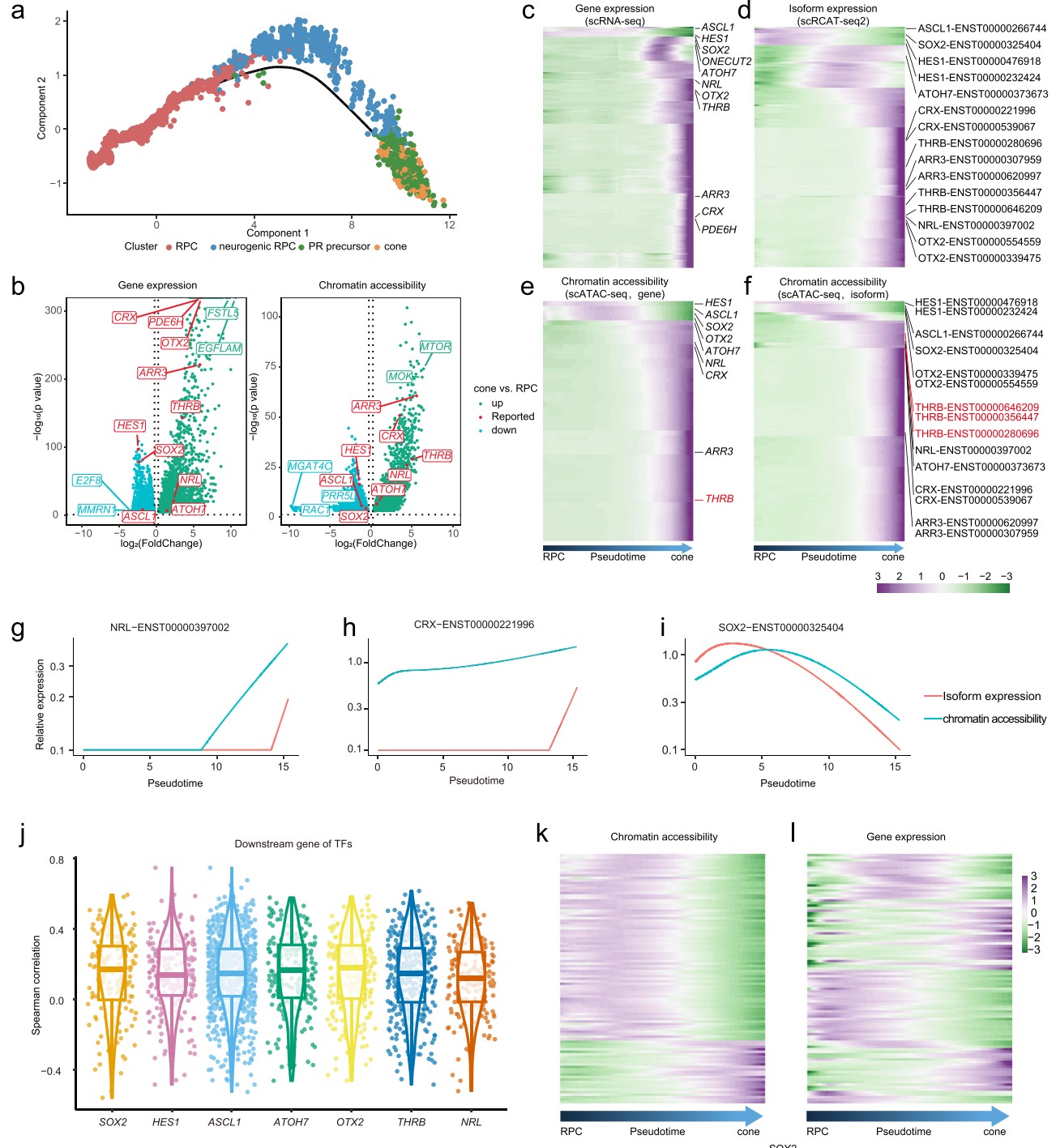

**Fig. 4 | Chromatin accessibility correlates with RNA transcription of fate determining factors of RPC. a** Trajectory map of RPC-cone development generated by pseudotime analysis of RPC, neurogenic RPC, PR precursor, and cone data. **b** Volcano plot showing genes with changes in expression (left) and chromatin accessibility (right) between RPCs and cones. *P*-values were calculated using a two-sided Wilcoxon rank sum test. **c**–**f** Heatmap of dynamic gene (**c**) and isoform (**d**) during RPC-cone development. Heatmap of dynamic chromatin accessibility at gene (**e**) and isoform (**f**) promoters during RPC-cone development. **g**–**i** Changes in the relationship between promoter accessibility and expression of key genes NRL (**g**), CRX (**h**), and SOX2 (**i**) main isoforms during RPC differentiation into cones. **j** Scatter plot showing the correlation between chromatin accessibility of downstream target genes of key transcription factors and gene expression. The boxplot shows the median as center line, the interquartile range (IQR) as a box, the whiskers indicate 1.5 × IQR. **k-l** Relationship between chromatin accessibility and gene expression of downstream target genes regulated by transcription factor SOX2 along the RPC differentiation trajectory. Source data are provided as a Source Data file.

is dependent on the genome reference to determine the differential structures among transcripts. This means the method is limited by the completeness and accuracy of the reference genome annotation. Second, the sample size in this study was limited, which restricts our ability to fully reveal the functions of the identified transcripts. We plan to validate the functions of the undiscovered transcripts, particularly those of fate-determining factors, in future studies with larger sample sizes.

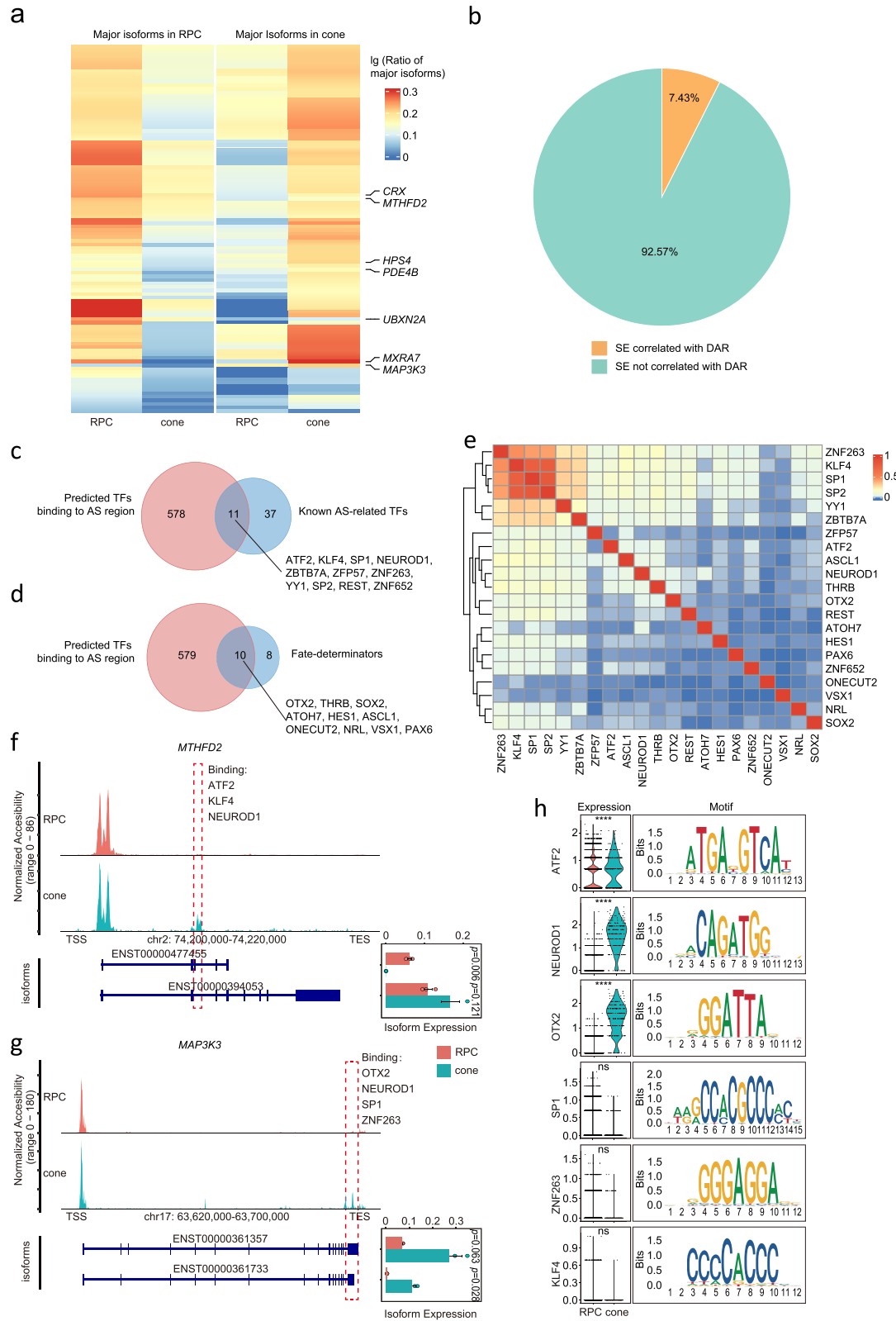

**Fig. 5 | Correlation of RNA isoform choice with chromatin accessibility and TFs during RPC development. a** Heatmaps of the major isoform choices in RPCs and cones. **b** Pie chart showing the ratio of isoform alternative splicing events (SE) correlated with differentially accessible region (DAR), $n = 2292$. **c** Venn diagram showing the overlaps between predicted TFs binding to alternative splicing (AS) region and known AS-related TFs. **d** Venn diagram showing the overlaps between predicted TFs binding to alternative splicing (AS) region and fate determinators. **e** Co-localization analysis of TFs related to splicing and fate determination at splicing sites in the diagrams (**c**, **d**) ($n = 21$). **f-g** Schematic representation of MTHFD2 (**f**) and MAP3K3 (**g**) locus undergoing SE with concurrent changes in chromatin accessibility. Significance was computed using two-sided $t$-test. Data are presented as mean values +/- SEM ($n = 3$ duplicate samples). **h** Integration of transcription factor expression changes and motif analysis at variable splicing sites in *MTHFD2* and *MAP3K3*. Source data are provided as a Source Data file.

## Methods

### Culture of cell lines

Human embryonic stem cell (hESC) line H9 was kindly provided by the Stem Cell Bank of the Chinese Academy of Sciences. IRB approval was obtained from the Zhongshan Ophthalmic Center, Sun Yat-sen University (2024KYPJ081). HESC were cultured in Essential 8 medium, and grown on 6-well plates coated with Vitronectin (VTN-N). Upon achieving >80% confluence, these cells were passaged using Versene and routinely divided twice a week. HEK293T cells (ATCC No. CRL-3216) were cultivated in a culture medium composed of DMEM enriched with 5% fetal bovine serum (FBS), nonessential amino acids, and penicillin-streptomycin. For subculturing, the cells were treated with 0.25% Trypsin/EDTA when they reached 80% confluency.

### Constructing the scRCAT-seq2 library and sequencing in low throughput

For the low-throughput scRCAT-seq2, we primarily utilize it for testing hESC and HEK293T cells. The scRCAT-seq2 library construction involves several steps, including mRNA capture, cDNA synthesis, single-molecule self-circle formation, rolling-circle amplification, cDNA fragmentation, TSS and TES library construction, Illumina platform sequencing, and data analysis.

Firstly, the mRNA capture and cDNA synthesis steps are performed. After obtaining the single-cell suspension, the Lysis Buffer (0.4 µL 50% Poly-ethylene Glycol 8000, 0.3 µL 2% Triton-X100, 0.04 µL RNase inhibitor (40 U/µL), 0.2 µL dNTPs (10 mM), 0.02 µL Oligo(dT) primer (100 µM), 0.5 µL SIRV-Set3 (1:10000), 1.54 µL Nuclease Free Water), RT Mix (0.1 µL Tris-HCl pH8.3 (1 M), 0.12 µL NaCl (1 M), 0.1 µL MgCl$_2$ (100 mM), 0.04 µL GTP (100 mM), 0.32 µL DTT (100 mM), 0.05 µL RNase inhibitor (40 U/µL), 0.08 µL TSO primer (100 µM), 0.04 µL Maxima H-minus RT, 0.15 µL Nuclease Free Water), and PCR Mix (5 µL KAPA HiFi HotStart ReadyMix (2x), 0.25 µL oligo primer (10 µM), 0.25 µL TSO ISPCR primer (10 µM), 0.5 µL Nuclease Free Water) are prepared. Then, pick up single cells through an oral pipette into PCR tubes containing 3 µL Lysis Buffer. After shaking and brief centrifugation, the tubes are incubated in a PCR machine: 72 °C for 10 min. The tubes are quickly placed on ice, and then 1 µL RT Mix is added. The reaction proceeds as follows: 42 °C for 90 min, 10 cycles of 50 °C for 2 min and 42 °C for 2 min, and 70 °C for 15 min. The resulting mixture is purified using 1x AMPure XP beads and eluted with 8.5 µL Nuclease Free Water. Next, 1 µL 10x ExoI Buffer and 0.5 µL ExoI (20 U/µL) are mixed with the purified mixture. The reaction proceeds as follows: 37 °C for 30 min, 80 °C for 10 min. After the reaction, the product is purified using 1x AMPure XP beads and eluted with 4 µL Nuclease Free Water. Then, 6 µL PCR Mix is added, and the reaction proceeds as follows: 98 °C for 3 min, 18–25 cycles of 98 °C for 20 s, 65 °C for 30 s, and 72 °C for 4 min, followed by 72 °C for 5 min.

Next, cDNA single-molecule self-circle formation is performed. 200 ng of cDNA is mixed with 0.04 µM of linker primer and 6 µL 2x NEBuilder HiFi DNA Assembly Master Mix to achieve a final volume of 12 µL. The reaction is incubated at 55 °C for 60 min. Then, 1 µL Exonuclease I, 1 µL Exonuclease III: Lambda Exonuclease (1:10), and 2 µL 10x NEB Buffer 1 are added to the reaction, resulting in a final volume of 20 µL. The reaction is digested at 37 °C for 30 min and then heated at 80 °C for 10 min. The product is purified using 1.8x AMPure XP beads and eluted with 10 µL Nuclease Free Water.

Rolling circle amplification is then performed. The circularized cDNA is mixed with 5 µL phi29 DNA Polymerase Reaction Buffer, 2 µL dNTP (2 mM), 1 µL Recombinant Albumin (2 mg/mL), and 1 µL Phi29 primer (10 µM), and the volume is supplemented with Nuclease Free Water to reach 19 µL. The reaction proceeds as follows: 95 °C for 5 min, immediately followed by incubation on ice for 2 min. Then, 1 µL phi29 DNA Polymerase is added, and the mixture is incubated at 37 °C for 16 hours, followed by 65 °C for 10 min. Finally, the product is purified using 0.6x AMPure XP beads and eluted with 10 µL Nuclease Free Water to obtain linear, multi-copy cDNA.

cDNA fragmentation is performed using 50 ng of linear, multi-copy cDNA. The cDNA is mixed with 2 µL 5x Tagment Buffer L, 0.4 µL TTE Mix (4 pmol/µL), and Nuclease Free Water to achieve a final volume of 10 µL. The reaction is incubated at 55 °C for 10 min. The product is purified using 1x AMPure XP beads and eluted with 20 µL Nuclease Free Water to obtain fragmented cDNA.

TSS and TES library construction are performed separately using 20 µL of fragmented cDNA. For TSS library construction: 10 µL of fragmented cDNA is mixed with 1 µL Tn5-i5 primer (10 µM), 1 µL anti-tag1 primer (10 µM), 15 µL KAPA HiFi HotStart ReadyMix (2x), and Nuclease Free Water to reach a final volume of 30 µL. The reaction proceeds as follows: 98 °C for 3 min, 6 cycles of 98 °C for 20 s, 60 °C for 20 s, and 72 °C for 90 s, followed by 72 °C for 5 min. After purification using 1x AMPure XP beads, the resulting product is eluted with 10 µL Nuclease Free Water, yielding TSS PCR1 product. Then, the TSS PCR1 product is mixed with 1 µL Tn5-i5 primer (10 µM), 1 µL i7-anti-tag1 primer (10 µM), 15 µL KAPA HiFi HotStart ReadyMix (2x), and Nuclease Free Water to reach a final volume of 30 µL. The reaction proceeds as follows: 98 °C for 3 min, 5 cycles of 98 °C for 20 s, 60 °C for 20 s, and 72 °C for 90 s, followed by 72 °C for 5 min. After purification using 1x AMPure XP beads, the resulting product is eluted with 10 µL Nuclease Free Water, yielding TSS PCR2 product. Next, the Nextera XT DNA Sample Preparation Kit (Illumina) is used for tagging. Specifically, the TSS PCR2 product is mixed with 1 µL of 10 µM Index 1 primer (N7), 1 µL of 10 µM Index 2 primer (N5), 15 µL KAPA HiFi HotStart ReadyMix (2x), and Nuclease Free Water to reach a final volume of 30 µL. The reaction proceeds as follows: 98 °C for 3 min, 4–10 cycles of 98 °C for 20 s, 55 °C for 20 s, and 72 °C for 90 s, followed by 72 °C for 5 min. After purification using 0.55x AMPure XP beads, the resulting product is eluted with 20 µL Nuclease Free Water, generating long fragment library. Furthermore, purification using 0.55x-0.9x AMPure XP beads is performed, and the product is eluted with 20 µL Nuclease Free Water, yielding short fragment library. The TES library construction method is similar to TSS, except that the anti-tag1 primer and i7-anti-tag1 primer in TSS are replaced with anti-TSO primer and i7-anti-TSO primer. Finally, the library is assessed using an Agilent Bioanalyzer 2100 or Qsep100™ automated nucleic acid and protein analysis system, and sequencing is performed on the Illumina NovaSeq 6000 platform using the PE150 model. For primer sequence information and details of the reagents used, please refer to the Supplementary Data 2 and Data 3.

### Differentiation of human retinal organoids

In this study, we cultured hESCs in Essential 8 (E8) medium on Vitronectin (VTN-N)-coated plates. To induce retinal differentiation, we dissociated hESC colonies using dispase and allowed them to form small cell clusters. Gradually, the medium was switched from E8 to neural induction medium (NIM) over a period of four days. The NIM contained DMEM/F12(1:1), N2 supplement (1%), MEM nonessential amino acids (1%), penicillin-streptomycin(1%), and heparin sulfate(2 mg/mL).To enhance the efficiency of retinal differentiation, we added recombinant human BMP4(50 ng/mL) starting from day 6. Every third day, we replaced half of the NIM volume with fresh NIM containing BMP4. On day 7, the cell aggregates were reseeded onto VTN-N-coated plates using NIM medium supplemented with 10% fetal bovine serum (FBS) and BMP4. On day 16, we manually dislodged the neural rosettes from the plates and maintained them in retinal differentiation medium (RDM) for the formation of retinal organoids. The RDM consisted of DMEM/F12 (3:1), B27 supplement (2%), MEM nonessential amino acids (2%), and penicillin-streptomycin (2%). Starting from day 30, we supplemented the RDM medium with 10% FBS, 100 µM taurine, 2 mM GlutaMAX, and 0.5 µM retinoic acid to support the long-term culture of retinal organoids[41].

## scRCAT-seq2 library construction for D45 human retinal organoids

On day 45, we dissociated these organoids into single-cell suspensions by treating them with Accutase at 37 °C for 30 minutes. The suspensions were then filtered through a 35-μm cell strainer, and the cells were resuspended in PBS containing 0.04% bovine serum albumin. Then, we used the Chromium Next GEM Single Cell Multiome ATAC + Gene Expression kit from 10x Genomics, following the manufacturer's protocol to generate scRNA-seq and scATAC-seq libraries. Preserving cDNA samples of scRNA- seq for high-throughput scRCAT-seq2 library construction.

## Constructing the scRCAT-seq2 library and sequencing in high throughput

First, the construction and sequencing of multiomics libraries for scRNA-seq and scATAC-seq were carried out using the Chromium Next GEM Single Cell Multiome ATAC + Gene Expression kit from 10x Genomics. Subsequently, the cDNA obtained from this process was used for the construction and sequencing of the scRCAT-seq2 library.

The main steps of scRCAT-seq2 are as follows: cDNA retrieval, single-molecule self-ligation, rolling circle amplification, random fragmentation, TSS and TES library construction, sequencing on the MGISEQ-2000 platform, and data analysis.

The first step is the retrieval of cDNA. We utilized the Chromium Next GEM Single Cell Multiome ATAC + Gene Expression kit from 10x Genomics to construct multiomics libraries for scRNA-seq and scATAC-seq and obtain the cDNA. If there is insufficient cDNA, PCR amplification is performed to obtain a sufficient amount of cDNA. The specific procedure involves taking 10 μL of cDNA and adding 1 μL 10x-Read1 primer(10 μM), 1 μL 10x-TSO primer(10 μM), 15 μL KAPA HiFi HotStart ReadyMix (2×), and supplementing with Nuclease Free Water to a total volume of 30 μL. The program is run as follows: 98 °C for 3 min, (5-12 cycles: 98 °C for 20 s, 65 °C for 20 s, 72 °C for 2 min), and a final step at 72 °C for 5 min. After purification with 0.6x AMPure XP beads, the cDNA is eluted with 20 μL Nuclease Free Water to obtain sufficient cDNA products.

The cDNA is then self-ligated into single molecules. 200 ng of cDNA is mixed with 0.04 μM of 10x-linker primer and 6 μL 2x NEBuilder HiFi DNA Assembly Master Mix to a final volume of 12 μL. After incubating at 55 °C for 60 min, 1 μL Exonuclease I, 1 μL Exonuclease III: Lambda Exonuclease (1:10), and 2 μL 10x NEB Buffer1 are added to the reaction mixture to a final volume of 20 μL. The mixture is digested at 37 °C for 30 min and then incubated at 80 °C for 10 min. After the reaction is completed, the cDNA is purified using 0.8x AMPure XP beads and eluted with 10 μL Nuclease Free Water.

Next, the circularized cDNA is amplified through rolling circle amplification. The circularized cDNA is added to 2 μL phi29 DNA Polymerase Reaction Buffer, 0.4 μL dNTP (10 mM), 1 μL Recombinant Albumin (2 mg/mL), 1 μL 10x-Phi29 primer (100 μM), and Nuclease Free Water to a total volume of 19 μL. The reaction is performed at 95 °C for 5 min followed by immediate incubation on ice for 2 min. Then, 1 μL phi29 DNA Polymerase is added, and the mixture is thoroughly mixed and incubated at 37 °C for 16 h, followed by an additional incubation at 65 °C for 10 min. Finally, the cDNA is purified using 0.6x AMPure XP beads and eluted with 10 μL Nuclease Free Water to obtain linearly amplified cDNA.

The next step is fragmentation of the cDNA. 50 ng of linearly amplified cDNA is mixed with 2 μL 5 × Tagment Buffer L, 0.4 μL TTE Mix (4 pmol/μL), and Nuclease Free Water to a total volume of 10 μL. The reaction is performed at 55 °C for 10 min, followed by purification using 1x AMPure XP beads. The fragmented cDNA is then eluted with 20 μL Nuclease Free Water to obtain fragmented cDNA products.

The fragmented cDNA products are divided into two portions for TSS (transcription start site) and TES (transcription end site) library construction. For TSS library construction, 10 μL fragmented cDNA is mixed with 1 μL Tn5-i7 primer(10 μM), 1 μL 10x-polyA(10 μM), 15 μL KAPA HiFi HotStart ReadyMix (2×), and supplemented with Nuclease Free Water to a total volume of 30 μL. The program is run as follows: 98 °C for 3 min, (6 cycles: 98 °C for 20 s, 60 °C for 20 s, 72 °C for 90 s), and a final step at 72 °C for 5 min. After purification with 1x AMPure XP beads, the cDNA is eluted with 10 μL Nuclease Free Water to obtain TSS PCR1 products. The obtained PCR1 products incubated at 80 °C for 10 min. After the reaction is completed, the cDNA is purified using 0.8x AMPure XP beads and eluted with 10 μL Nuclease Free Water.

Next, the circularized cDNA is amplified by rolling circle amplification. The circularized cDNA is added to 2 μL phi29 DNA Polymerase Reaction Buffer, 0.4 μL dNTP (10 mM), 1 μL Recombinant Albumin (2 mg/mL), and 1 μL 10x-Phi29 primer (100 μM), and supplemented with Nuclease Free Water to a total volume of 19 μL. The reaction is initiated at 95 °C for 5 min, followed by incubation on ice for 2 min. Then, 1 μL phi29 DNA Polymerase is added, and the mixture is incubated at 37 °C for 16 hours, followed by 65 °C for 10 min. Finally, the amplified cDNA is purified using 0.6x AMPure XP beads and eluted with 10 μL Nuclease Free Water to obtain linear multiple-copy cDNA.

The cDNA is then fragmented. 50 ng of linear multiple-copy cDNA is combined with 2 μL 5 × Tagment Buffer L and 0.4 μL TTE Mix (4 pmol/μL), and supplemented with Nuclease Free Water to a total volume of 10 μL. The reaction is performed at 55 °C for 10 min. After purification with 1x AMPure XP beads, the cDNA is eluted with 20 μL Nuclease Free Water to obtain fragmented cDNA products.

TSS and TES library construction. The obtained 20 μL fragmented cDNA product is divided into two parts for the construction of TSS and TES libraries, respectively. TSS library construction: Take 10 μL of fragmented cDNA product and add 1 μL Tn5-i7 primer (10 μM), 1 μL 10x-polyA (10 μM), 15 μL KAPA HiFi HotStart ReadyMix (2×), and supplement with Nuclease Free Water to a total volume of 30 μL. The program is run as follows: 98 °C for 3 min, (6 cycles: 98 °C for 20 s, 60 °C for 20 s, 72 °C for 90 s), and a final step at 72 °C for 5 min. After purification with 1x AMPure XP beads, the cDNA is eluted with 10 μL Nuclease Free Water to obtain the TSS PCR1 product. The obtained PCR1 product is added to 1 μL Tn5-i7 primer (10 μM), 1 μL 10x-i5-polyA(10 μM), 15 μL KAPA HiFi HotStart ReadyMix (2×), and supplemented with Nuclease Free Water to a total volume of 30 μL. The program is run as follows: 98 °C for 3 min, (5 cycles: 98 °C for 20 s, 60 °C for 20 s, 72 °C for 90 s), and a final step at 72 °C for 5 min. After purification with 1x AMPure XP beads, the cDNA is eluted with 10 μL of Nuclease Free Water to obtain the TSS PCR2 product. Next, labeling is performed using the Nextera XT DNA Sample Preparation Kit (Illumina). The specific procedure involves adding the obtained PCR2 product to 1 μL of 10 μM Index 1 primer(N7), 1 μL of 10 μM Index 2 primer(N5), 15 μL of KAPA HiFi HotStart ReadyMix (2×), and supplementation with Nuclease Free Water to a total volume of 30 μL. The program is run as follows: 98 °C for 3 min, (4-10 cycles: 98 °C for 20 s, 55 °C for 20 s, 72 °C for 90 s), and a final step at 72 °C for 5 min. After purification with 0.55x AMPure XP beads, the cDNA is eluted with 20 μL of Nuclease Free Water to obtain the large fragment library product. After purification with 0.55x-0.9x AMPure XP beads, the cDNA is eluted with 20 μL of Nuclease Free Water to obtain the short fragment library product. The TES library construction method is the same as TSS, except that Tn5-i7 primer, 10x-polyA and 10x-i5-polyA in TSS are replaced with Tn5-i5 primer, 10x-anti-TSO and 10x-i7-anti-TSO, respectively. Finally, the library is evaluated using the Agilent Bioanalyzer 2100 or Qsep100™ automated nucleic acid and protein analysis system, and sequencing is performed on the MGI MGISEQ-2000 platform using the PE150 model. The primer sequence information can be found in the Supplementary Data 2.

## scRCAT-seq2 was used to test frozen kidney tissues in mice

The experiment was conducted on C57BL/6 mice aged 6–8 weeks. All animal procedures complied with relevant ethical regulations for

animal experimentation and research, and were approved by the Institutional Animal Care and Use Committee (IACUC) of Sun Yat-sen University Zhongshan Ophthalmic Center (Z2021078). Mice were maintained under standard conditions (12 h light-dark cycle, with adequate food and water). To obtain kidney tissues, mice were euthanized by cervical dislocation, dissected, and kidney tissues were obtained. Kidney tissue blocks were flash-frozen in liquid nitrogen and stored long-term at −80 °C. The main steps of testing frozen tissues with scRCAT-seq2 involved obtaining single-cell nuclei from frozen kidney tissues through grinding, followed by library construction and sequencing analysis using the low-throughput scRCAT-seq2 method. Finally, evaluation was done based on cDNA detection results and the number of genes and isoforms obtained from each cell.

## scRCAT-seq2 was used to test monkey corneal limbal epithelium cells

A 19 year-old macaque monkey (*Macaca fascicularis*) was sourced from the Huazhen Laboratory Animal Breeding Centre in Guangzhou, China. The animal was housed at a constant temperature of 25 °C, following a 12 h light and 12 h dark cycle at the Huazhen Laboratory Animal Breeding Centre. All animal procedures were carried out in alignment with the Principles for the Ethical Treatment of Non-Human Primates and the Statement for the Use of Animals in Ophthalmic and Vision Research, and received the approval of the Institutional Animal Care and Use Committee (IACUC) at Zhongshan Ophthalmic Center, Sun Yat-sen University (2019–150).

After the animals were anesthetized, the corneas were excised and subjected to digestion in a Dispase II solution (10 mg/mL, Sigma, SCM133) for 2 hours at 37°C. Subsequently, the corneal limbal epithelium was meticulously dissected with forceps and moved into a 1.5 mL EP tube, where it was treated with a Dispase II solution (2.5 mg/mL) at 37°C for an additional 15 min. During this period, the cell aggregates were disaggregated into single cells with gentle shaking every 5 minutes. Then, we used the Chromium Next GEM Single Cell 5′ Reagent Kits v2 from 10x Genomics, following the manufacturer's protocol to generate scRNA-seq library. Finally, cDNA samples from the scRNA-seq were preserved for the construction of the high-throughput scRCAT-seq2 library.

## Read alignments and gene expression estimation

Raw non-demultiplexed paired-end fastq files were first preprocessed to retain reads containing the complete structure at TSS and TES. These reads were then mapped to the reference genome using the STARSOLO procedure with STAR (v2.7.10b) for both TSS and TES data, generating a gene expression matrix. To extract and identify reads containing the CB and UMI during preprocessing, we searched for the following pattern: oligo(AACGCAGAGT) + CB + UMI + tag(AGTTAACGCT) for both TSS and TES sequencing data. We then identified the cDNA sequences following the TSO (GCAGGGTTGGG) and the polyA sequences for TSS and TES data, respectively. Due to differences in sequence orientation, we reverse complemented the sequence oligo(AACGCAGAGT) + CB + UMI + tag(AGTTAACGCT) in the raw fastq data to match the sequence in the TES data. Subsequently, we aligned the fastq files to the reference genome using STAR with the following parameters: --soloType CB_UMI_Simple --soloCBstart 11 --soloCBlen 6 --soloUMIstart 17 --soloUMIlen 12 --outSAMattributes NH HI nM AS CR UR CB UB GX GN. The resulting bam files were used for further downstream analysis.

If using the 10x Genomics high-throughput platform, the oligo(AACGCAGAGT) + CB + UMI + tag (AGTTAACGCT) and TSO (GCAGGGTTGGG) should be modified as read1 (CTTCCGATCT) + CB + UMI + tag (TTTTTTTTTTT) and TSO (GAGTACATGGG), respectively.

The reference and gene annotation used in this study as follows: human: hg38 and GENCODE v44 from GENCODE, mouse: mm39 and GENCODE vM33 from GENCODE, monkey: GCA_012559485.3 and GCF_012559485.2(v102) from NCBI.

## Isoform identification

To identify the isoforms, we first merged the data derived from the TSS and TES libraries using Samtools, following the read mapping with STAR. We then collapsed the reads with the same unique molecular identifier (UMI) that mapped to the same gene, and extracted their aligned genomic coordinates. Next, we matched all the features, including TSS, TES, and intron and exon boundaries, with the features of annotated isoforms. We then assigned the UMIs to the corresponding isoforms based on the overlapping genomic features. Finally, for the UMIs that did not correspond to reads covering specific features of an isoform, we assigned them by estimating the maximal probability of the isoforms the reads belong to.

## Data processing of ScISOr-seq data

The raw data obtained after SMRT sequencing was processed using the official SMRT software SMRT-Link and CCS for conversion and filtering to produced circular consensus reads (CCS data). Aligning the CCS data to the human genome (hg38) using minimap2 (version 2.17), the resulting alignments were further compared to hg38 transcriptome annotation using gffcompare (version 0.11.2) and collapse to remove redundancy.

## Data analysis of scRICA-seq for the D45 human retinal organoids

The D45 human retinal organoids multimodal raw sequencing data was aligned to the human genome (hg38) using Cell Ranger (version 6.1.1), enabling analysis of gene expression and chromatin accessibility. Further analysis was conducted using the R packages Seurat (version 4.1.0) and Signac (version 1.6.0). Initially, a Seurat object was created based on the gene expression count matrix. Then, an ATAC-seq peak-count matrix was added as a Chromatin Assay to the Seurat object. Standardized methods were applied independently to preprocess and reduce the dimensionality of the two assays. For the Seurat assay, the data was normalized, and the "RunPCA" function was used for linear dimensionality reduction, selecting the top 50 principal components (1:50 dims) as parameters. Following that, the "RunUMAP" function was employed for nonlinear dimensionality reduction. Regarding the Chromatin assay, after normalizing the data, linear dimensionality reduction analysis using the "RunSVD" function was performed followed by nonlinear dimensionality reduction analysis using the "RunUMAP" function. The "RunUMAP" function was utilized with the parameters reduction = 'lsi' and dims = 2:50 to exclude the first dimension, which is typically correlated with sequencing depth. Afterwards, the "FindMultiModalNeighbors" function was employed to calculate the weighted nearest neighbor (WNN) graph, which represents the weighted combination of the RNA-seq and ATAC-seq modalities. This graph was subsequently used for UMAP visualization and clustering. The "FindAllMarkers" function was employed to identify marker genes for each cluster within the WNN object, with criteria of Log2FC > 0.25 and min.pct > 0.25. By comparing the identified markers with known early retinal development markers reported in the literature, cell type annotations were assigned to each cluster.

## Inference of pseudotime and trajectory

Monocle (version 2.27.0) was applied to unravel the lineage differentiation trajectories of early retinal cells. Firstly, a monocle object was created using the "as.CellDataSet" function. The "reduceDimension" function was applied to perform dimensionality reduction analysis using the DDRTree method, followed by pseudotime analysis using the "orderCells" function.

## Data analysis of differential expression and differential accessibility

Differential expression and accessibility analysis between two cell types were performed using the "FindMarkers" function from the R package Seurat. The Wilcoxon rank-sum test was used to identify differentially expressed genes ($|\log_2(\text{fold change})| > 0.25$, adjusted $p$-value $< 0.05$), while a logistic regression framework was employed to determine differentially accessible peaks ($|\log_2(\text{fold change})| > 0.1$, $p$-value $< 0.0005$). Finally, a scatter plot will be generated using Pearson correlation to depict the correlation between the $\log_2(\text{fold change})$ values of expression and accessibility differences for genes that exhibit both significant differential expression and promoter region differential accessibility. A linear regression with a 95% confidence interval will also be added to the plot.

## Peak annotation

We used the R package ChIPseeker[42] to map peaks to the nearest genes and gene regions. Promoter sites were defined as the TSS (Transcription Start Site) upstream 2000 bp and downstream 200 bp.

## Data analysis of alternative splicing

First, for the isoform expression data identified and quantified by the scRCAT-seq2 program, the functions "NormalizeData" and "AverageExpression" form R package Seurat were used to calculate the normalized average expression levels of each isoform in each cell type. The processed isoform expression quantification data and genome annotation were imported into the R package IsoformSwitchAnalyzeR(version 4.3.2), and the function "isoformSwitchTestDEXSeq" was used to identify isoforms that significantly switch between two cell types (alpha = 0.05, dIFcutoff = 0.1). Subsequently, the function "isoformsanalyzeAlternativeSplicing" was employed to statistically analyze alternative splicing events among isoforms with significantly altered proportions between the two cell types.

## Identification of TFs binding to alternative splicing

To identify TFs binding to alternative splicing, motif analysis was performed on accessible peaks that overlap with the regions 100 bp upstream and downstream of alternative splicing sites in the D45 human retinal organoids multimodal data. Firstly, the "CreateMotifObject" function from the R package Signac was utilized to generate a motif object, which was then stored within the multimodal WNN object. Additionally, after calculating sequence characteristics of peaks, the "FindMotifs" function from the R package chromVAR (version 1.16.0) was used to enrich motifs related to known transcription factor binding sites. In further detail, the R package motifmatchr(version 4.3.2) was used to accurately retrieve motifs matched at the peaks associated with alternative splicing, and these motifs were filtered using a $p$-value cutoff of $< 0.00005$ and a score threshold of $> 10$.

## Statistics and reproducibility

Statistical details for each experiment are described in the figure legends. For testing scRCAT-seq2 with hESC and HEK293T cells, six individual cells from each cell type were separately subjected to scRCAT-seq2 experiments to serve as six replicate samples. The applicability of scRCAT-seq2 to frozen tissues was validated using 20 replicate experiments with mouse kidney cells. No statistical method was used to predetermine sample size. No data were excluded from the analyses. The experiments were not randomized. The Investigators were blinded to allocation during experiments and outcome assessment.

## Reporting summary

Further information on research design is available in the Nature Portfolio Reporting Summary linked to this article.

## Data availability

All sequencing data generated in this study has been uploaded at Gene Expression Omnibus (GEO) with accession number: GSE251754. Go to https://www.ncbi.nlm.nih.gov/geo/query/acc.cgi?acc=GSE251754. Source data are provided with this paper.

## Code availability

The code used in this study is available from GitHub (https://github.com/huyoujinlab/scRCAT-seq2).

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

## Acknowledgements

Y.H. gratefully acknowledges support from National Key R&D Program of China (2023YFC2506100), the National Natural Science Foundation of China (32171445, 82471128), Key-Area Research and Development Program of Guangdong Province (NO: 2023B1111020006), and the Science and Technology Program of Guangzhou (202201020624, SL2024A03J01243).

## Author contributions

Y.H., and S.Z. conceived and designed the study. S.Z., Y.X., Y.Q., Z.C., X.L., X.C., W.D. and J.C. performed experiments. Y.H., X.M., S.Z., Y.X., J.Z. and X.J. analyzed the data and performed statistical analyses. Y.H., G.F., S.Z., X.M. and Y.X. interpreted the data and wrote the manuscript in discussion with all authors.

## Competing interests

The authors declare no competing interests.
