## [Peer Review File · Nature Communications]

Simultaneous profiling of RNA isoforms and chromatin accessibility of single cells of human retinal organoidsReviewers' Comments:

Reviewer #1:

Remarks to the Author:

In this article, the authors address the limitations of current single-cell multi-omics sequencing. The existing approaches, mainly based on second-generation short-read sequencing, have inefficiencies in detecting RNA splicing heterogeneity. While long-length sequencing can analyze RNA splicing, it suffers from low throughput and limited detection of transcripts at the single-cell level, with a notable absence of single-cell profiling of chromatin accessibility. The study introduces a new single-cell multi-omics assay using short-read sequencing, capable of simultaneously detecting single-cell full-length RNA isoforms and DNA accessibility. This method achieved an impressive 94.5% accuracy in resolving RNA transcript isoform and demonstrates twice the sensitivity in detecting single-cell gene expression compared to third-generation sequencing. Moreover, it can process over 10,000 single nuclei in a single run, facilitating high-throughput integrated analysis of single-cell RNA isoforms and DNA accessibility. The authors applied this method to construct a comprehensive cell atlas of human retinal organoids, revealing associations between isoform expression and chromatin accessibility of fate-determining factors. Overall, this innovative approach offers a promising technical solution for unraveling the intricate molecular mechanisms underlying gene expression regulation.

Major comments

It is yet unclear how the chromatin accessibility data can be helpful in understanding RNA splicing, which is the major new component of scRCAT-seq2 compared to scRCAT-seq. This type of result might significantly increase reader's interest in this new technology. For instance, what factors might regulate the detected RNA splicing events? Is the expression change of these factors associated with the change of the RNA splicing events, and does the change of chromatin accessibility at some sites explain the expression change of these factors?

Line 158-159, does the chromatin accessibility change at the many binding sites of these transcription factors in the genome? And does the expression of target genes change accordingly?

Line 111-113, how do the numbers change with sequencing depth or cost?

Line 142-148 and line 158-159, is it possible to show the genome browser tracks of the individual example genes to better visualize the RNA and ATAC read density.

Line 142-148 and line 158-159, is there any evidence suggesting that the changes of chromatin accessibility are required for the isoform expression changes?

The technology has been tested on fresh organoid samples. Does it also work on frozen tissues or FFPE samples?

Fig. 3e, is the heatmap plotted for individual genes or individual isoforms?

Minor comments

Replacing the RNA structure with a more specific phrase, such as the RNA isoform, might be better. RNA structure can have multiple meanings, such as the RNA secondary or 3D structure.

It might be better to replace the phrase epigenetic modification with chromatin accessibility and a more specific phrase.

Line 99, the verb should not be in the past tense.

Add a reference to the ScISOR-seq when first mentioned in line 106.

Remove the word "and" to break a long sentence into shorter sentences when possible.

Add p values for Fig 2d and e.

Reviewer #2:

Remarks to the Author:

The authors have developed and report a new method called scRICA-seq (single cell RNA isoform and chromatin accessibility sequencing), which is combined with their previously developed method scRCAT-seq to predict the chromatin accessibility and RNA isoforms based on single-cell RNA sequencing. The authors used Spike-In RNA Variants (SIRVs) 3 to validate scRICA-seq accuracies for isoform predictions, and further applied human retinal organoids to support their functions in detecting various cell types with typical cell markers. The new method presents an alternative to several recently developed single-cell RNA isoform detection algorithms/tools and could have added value; however, further comparative validation is needed. I have the following concerns that need to be addressed for better assessment and validity of the algorithm/tool presented.

Major concerns:

1. As is imperative for all newly developed methods, cross comparisons are essential to make the new method valuable because developers always claim their tool is the most superior method. In the introduction and discussion sections, the authors repeatedly emphasize the deficiencies of short reads methods and third generation sequencing; it would help if the authors would provide more details and references for the deficiencies of the methods referenced. For example, third generation sequencing is expensive and it is unclear how much data needs to be generated and at what sensitivity level for this vs other algorithms to succeed comparatively.
2. Several methods with same or similar functions are mentioned in the manuscript, including, ScISOR-Seq, ATAC-seq/ ATAC-seq2. The authors should also include more recent methods developed in the last two years, including but not limited to HIT-scISOseq (PMID: 37149708) and Scasa (PMID: 34864849).
3. The accuracies were calculated based on SIRV-set 3 kits, which contains 69 isoforms. It is unclear why the authors chose this kit for their accuracy evaluation (and why only this kit). Some explanation is needed as this is a small RNA isoform set and accuracy is one of the most critical conditions of the new method.
4. When comparing the sensitivity to ScISOR-seq, the authors claimed that scRCAT-seq2 was demonstrated to detect a higher number of genes and isoforms. Please explicitly state which set/references are used in the comparisons.
5. ScISOR-seq seems the only method applied in the cross comparison, as suggested in the previous points, the authors should assess the accuracies, sensitivities, and the functional detections on their human retinal organoids RNA set if possible to compare to previously developed methods.

Minor concerns:

It appears some of the paragraphs were written by AI tools based on anti-AI writing scanning programs. For example, the first paragraph (line 216-221), fourth and fifth paragraphs (line 243-266) in the discussion section, and first paragraph of introduction section (line 45-54), have very high probabilities for AI writings. While I have nothing against such tools being used, the authors should modify the written language generated to avoid this being picked up.

Reviewer #3:

Remarks to the Author:

The authors developed a single-cell multi-omics assay based on short-read sequencing for the simultaneous detection of single-cell full-length RNA isoforms and DNA accessibility (scRICA-seq). The authors did experiments to validate and characterize the design and performed multiple experiments to study the correlation between choices of isoforms of multiple fate-determining factors and chromatin accessibility in different cell types. This approach is novel and potentially useful. However, the details of this approach and data analysis and explanation should be significantly improved to highlight the novelty and significance either in technology development or biological insights.

- 1) The author mentioned that they used two different strategies to co-profile mRNA and chromatin accessibility, one of them is using a 10x platform. The authors should provide more details about these protocols both using schematic figures and discussing details in the Methods section.
- 2) The authors compared their methods with ScISO-seq, can they also compare with Nanopore sequencing of single-cell transcriptomes (scCOLOR-seq, Nature Biotechnology, volume 39, pages1517–1520 (2021))?
- 3) How robust is scRICA-seq? The authors should provide results by comparing technical replicates.
- 4) What is the throughput of scRICA-seq? How many cells it can process in each experiment? How many cells did they process in Figure 3-5?
- 5) The authors developed scRCAT-seq2 using an optimized protocol of scRCAT-seq, and they should provide data showing how much improvement scRCAT-seq2 achieves.
- 6) Figures 3 and 4 are very confusing. What protocols did the authors use for scATAC-seq + scRNA-seq? What is the purpose of Figures 3 and 4?
- 7) Line 204-209, the authors did not provide the chromatin accessibility data in Extended Data Fig. 6e-j, how did they conclude that “we found that the majority of splicing changes were independent of chromatin accessibility”?
- 8) I don't quite understand Figure 5k and l, how to read these figures?
- 9) The data analysis in Figure 5 is too vague. For example, Line 212, “This finding suggests a potential relationship between chromatin accessibility and RNA splicing.”
- 10) The detailed code for data analysis is missing.

Reviewer #1

Major comments

(1) It is yet unclear how the chromatin accessibility data can be helpful in understanding RNA splicing, which is the major new component of scRCAT-seq2 compared to scRCAT-seq. This type of result might significantly increase reader's interest in this new technology. For instance, what factors might regulate the detected RNA splicing events? Is the expression change of these factors associated with the change of the RNA splicing events, and does the change of chromatin accessibility at some sites explain the expression change of these factors?

Response: Thank you very much for the constructive suggestions.

During the RPC-to-cone differentiation, we identified a total of 2292 differentially expressed isoforms (**Supplementary Fig. 15**), and transcription factor motif analysis obtained 589 transcription factors enriched at the variable splicing sites. Among them, we identified 10 transcription factors that have been previously shown to have regulatory functions in RNA splicing, including NEUROD1, SP1, and KLF4 (**Fig. 5c, Dudek et al., 2021**).

Furthermore, by associating these transcription factors with the chromatin accessibility at their promoters, we found that the expression levels of many fate-determining factors, including NEUROD1, HES1, and THRB, are significantly correlated with the chromatin accessibility at their promoters (**Supplementary Fig. 17a-d**). Additionally, through the dynamic expression analysis of splicing-related factors, we found that the upregulation of splicing-related regulators such as KDM5B is also associated with increased chromatin accessibility at their promoters (**Supplementary Fig. 17e-g**). These results suggest that chromatin accessibility may regulate the expression of fate-determining factors and splicing factors, thereby participating in cell fate determination through the RNA splicing of downstream genes.

On the other hand, we observed that the regions with increased chromatin accessibility near dynamic splicing sites were enriched with 92 transcription factors (**Supplementary Fig. 16**), while the regions with decreased chromatin accessibility were enriched with 57 transcription factors (**Supplementary Fig. 16b**). Some of these transcription factors showed co-enrichment, suggesting that chromatin accessibility and multiple transcription factors may cooperatively regulate differential RNA splicing (**Fig. 5h**). For example, the transcription factor NEUROD1, which has been previously reported to be associated with RNA splicing (Dudek et al., 2021), is related to the differential isoform choice of many genes in the retinal organoids, including MTHFD2 (**Fig. 5f**), MAP3K3 (**Fig. 5g**), MXRA7 (**Supplementary Fig. 15c**), and CRX (**Supplementary Fig. 15f**). Meanwhile, NEUROD1 co-localizes with several other splicing-related factors, such as KLF4, SP1, and ZNF263, at these chromatin accessibility-differential sites (**Fig. 5h**). We hypothesize that chromatin accessibility may affect the binding and recruitment of NEUROD1 and other splicing-related factors, thereby influencing RNA splicing and retinal cell fate determination.

In addition, we also observed differential isoform selection in regions where chromatin accessibility did not change. Through transcription factor enrichment analysis, we found that the differential expression of transcription factors involved in RNA splicing regulation, such as NEUROD1 and ATF2, is associated with the isoform differences (**Supplementary Fig. 15f, g**), suggesting that the expression differences of transcription factors themselves may also regulate RNA splicing, and the detailed mechanism deserves further investigation.

In summary, scRCAT-seq2 provides an unprecedented opportunity to integrate the analysis of chromatin accessibility with the dynamic changes in isoforms at the single-cell level. Previous studies have shown that chromatin accessibility plays a role in RNA splicing through affecting RNA transcription elongation rates, influencing the binding of transcription factors and the recruitment of splicing factors, as well as regulating the expression of transcription-related factors (Lydia et al., 2017). The novel splicing isoforms and related fate-determining factors identified in this study provide new clues for integrating epigenetics, RNA transcription, and splicing research to elucidate the

mechanisms underlying retinal neuronal development and fate determination.

(2) Line 158-159, does the chromatin accessibility change at the many binding sites of these transcription factors in the genome? And does the expression of target genes change accordingly?

Response: Thank you for sharing your valuable feedback. Based on your suggestions, we have further analyzed the changes in chromatin accessibility at the binding sites of key fate-determining factors. The results showed that the accessibility of these sites underwent significant changes, including both up-regulation and down-regulation.

Our analysis revealed that chromatin accessibility in the binding regions of some key fate-determining factors (such as SOX2, HES1, ASCL1, THRB, etc.) was generally positively correlated with the expression levels of their corresponding genes. Notably, the significance of this correlation varies, suggesting differential contribution of chromatin accessibility to the transcriptional regulation among these transcription factors (**Fig. 4k-l, Supplementary Fig.14**).

Of note, we also observed that the expression levels of some target genes were not correlated with the changes in accessibility, indicating that the transcriptional regulation of these transcription factors may also be influenced by other various factors.

Overall, these results demonstrate the dynamic changes in chromatin accessibility and its potential role in the transcriptional regulation mediated by the fate-determining factors. This provides new insights into the molecular mechanisms governing retinal neuronal development and fate determination.

(3) Line 111-113, how do the numbers change with sequencing depth or cost?

Response: Thank you for your constructive suggestion. Following your advice, we included additional data in the revised manuscript showing the variation in the number of isoforms detected by scRCAT-seq2 with varying sequencing costs (**Fig. 2g**). Our results indicate that the saturation sequencing cost per cell for scRCAT-seq2 is approximately \$3.30 (~4M reads per cell). Furthermore, we performed a comparative analysis of isoform detection across different sequencing techniques, including scRCAT-seq2, ScISOr-seq, and SCAN-seq2, at varying costs. This analysis revealed that scRCAT-seq2 detects the highest number of isoforms at equivalent costs. Specifically, compared to the PacBio-based ScISOr-seq method, scRCAT-seq2 detects significantly more isoforms ($12,695 \pm 348$ vs. $1,041 \pm 41.1$) at the same cost (**Fig.2h**). Similarly, for the high-throughput samples, scRCAT-seq2 detects more isoforms compared to HIT-scISO-seq (**Fig. 2i**).

When compared to the Nanopore-based SCAN-seq2 method, scRCAT-seq2 again shows superior performance in cost-equivalent conditions ($12,695 \pm 348$ vs. $8,566 \pm 123$) (**Fig. 2g**), and scRCAT-seq2 detects more isoforms than Nanopore sequencing on the 10X platform (**Fig. 2i**). Lastly, compared to the original scRCAT-seq, scRCAT-seq2 detects a significantly higher number of isoforms ($12,695 \pm 348$ vs. $2,152 \pm 130$) (**Fig. 2g**). Overall, we believe that scRCAT-seq2 demonstrates superior single-cell isoform detection compared to existing methodologies.

(4) Line 142-148 and line 158-159, is it possible to show the genome browser tracks of the individual example genes to better visualize the RNA and ATAC read density.

Response: Thank you for the constructive suggestions.

To better visualize the RNA and ATAC data, we performed cell-type-specific analysis of chromatin accessibility and gene expression, and selected several key transcriptional regulators, such as HES1, ATOH7, THRB, CRX, and ARR3, and presented their chromatin accessibility and expression levels across different cell types using the Genome browser tracks (**Fig. 3g; Supplementary Fig. 9**).

The results showed that the chromatin accessibility of these transcriptional regulators was highly correlated with their expression levels in the retinal cell types. For example, HES1, ATOH7, THRB, CRX, and ARR3 exhibited clear cell-type-specific patterns of chromatin accessibility and expression in the retinal organoids (**Fig. 3g**). For GAP43 and ISL1, we found that they were mainly expressed in RGCs (**Fig. 3g**), but also showed relatively high chromatin accessibility in non-RGC cell types. This suggests that the cell-type-specific expression of these two genes in RGCs may also be mediated by other transcriptional regulators.

We have included these results in more detail in the supplementary materials (**Fig. 3; Supplementary Fig. 9**). We hope that these new analyses and visualizations can better elucidate the relationship between chromatin accessibility and gene expression regulation.

(5) Line 142-148 and line 158-159, is there any evidence suggesting that the changes of chromatin accessibility are required for the isoform expression changes?

Response: Thank you for the constructive suggestions.

The new method developed in this study enabled parallel analysis of the dynamic changes in the expression of different isoforms of these fate-determining factors (**Fig. 3**) and chromatin accessibility of the specific promoters during human retinal development, revealing their correlation. The chromatin accessibility of a portion of isoform promoter regions increased prior to the expression changes of the corresponding isoforms, including NRL, ARR3, CRX, and THRB (**Fig. 4c-f, Supplementary Fig. 12**), suggesting that the changed chromatin accessibility set the promoters into a poised state before transcriptional activation (**Fig. 4g-i, Supplementary Fig. 13**). Previous studies have confirmed that the expression of NRL is regulated by H3K27 epigenetic modifications (Iida et al., 2015), and the function of H3K27 in regulating chromatin accessibility and poised state has been demonstrated in various tissues including the nervous system and tumors (Brown et al., 2014; Ciceri et al., 2024). Therefore, we hypothesize that the open chromatin accessibility is a necessary factor for the activation of genes such as NRL, THRB, and ARR3.

Meanwhile, the expression changes of isoforms for some genes, including HES1, SOX2, and ASCL1, were concordant with the changes in chromatin accessibility. In fact, epigenetic modifications at histone H3K9 and H4K12, which are closely related to chromatin accessibility levels, have also been shown to directly regulate the expression of HES1 (Ferreira et al., 2017). Therefore, we hypothesize that chromatin accessibility is a necessary factor for the expression of specific isoforms of various fate-determining factors such as NRL and ASCL1.

We have supplemented this part of the results in the discussion section of the manuscript and plan to validate and investigate the mechanisms in the future.

(6) The technology has been tested on fresh organoid samples. Does it also work on frozen tissues or FFPE samples?

Response: Thank you for the suggestion.

Accordingly, we have conducted experiments and reviewed relevant literature to demonstrate the feasibility of applying this method to frozen tissues or FFPE samples. For frozen tissues, we performed experimental tests on frozen mouse kidney samples. The results showed high-quality cDNA and high-sequencing quality, and recovered thousands of genes and isoforms. The similarity between the transcriptomic data of individual frozen single cells is comparable to that of fresh samples, indicating that this method can be applicable to frozen samples (**Supplementary Fig. 7**). In fact, existing commercial multi-omics kits (such as the 10xgenomics Single Cell Multiome ATAC + Gene Expression kit) are also compatible with frozen samples. By using these kits to generate cDNA and then applying the scRCAT-seq2 method for isoform analysis with the cDNA, the integrated analysis of ATAC and isoforms can be conveniently achieved for frozen samples.

For FFPE tissues, previous studies have shown that the RNA in FFPE samples is severely degraded, making it impossible to obtain full-length cDNA. Currently, the transcriptome sequencing of FFPE samples is mainly based on probe-targeted capture or random primer-based methods for library construction and sequencing, but these

methods cannot obtain full-length cDNA (Xu et al., 2023; Sounart et al., 2023). Therefore, we believe that scRCAT-seq2 is not currently applicable for FFPE samples. In summary, our method is applicable to frozen tissues, but not to FFPE samples. We have updated this information in the discussion section.

Fig. 1. cDNA fragment size distribution from fresh (left) and FFPE (right) sample (Xu et al., 2023; Sounart et al., 2023).

(7) Fig. 3e, is the heatmap plotted for individual genes or individual isoforms?

Response: Thank you for your comment. In the original manuscript, **Fig. 3e** displays the expression patterns of individual marker genes across different retinal cell types. To enhance clarity and help readers understand these heatmaps, we made the following modifications to **Fig. 3**:

1. We added a heatmap displaying the isoforms corresponding to the marker genes (**Fig. 3d**).
2. We included a title for each heatmap to indicate the data source.

Minor comments

Replacing the RNA structure with a more specific phrase, such as the RNA isoform, might be better. RNA structure can have multiple meanings, such as the RNA secondary or 3D structure.

Response: Thank you very much for your suggestion. In the revised manuscript, we have replaced all instances of "RNA structure" with "isoform" to avoid ambiguity.

It might be better to replace the phrase epigenetic modification with chromatin

accessibility and a more specific phrase.

Response: Thank you for your constructive suggestions. We have replaced "epigenetic modification" with "chromatin accessibility" in the appropriate places.

Line 99, the verb should not be in the past tense.

Response: Thank you for your constructive suggestion. We have corrected the verb tense in the manuscript.

Remove the word “and” to break a long sentence into shorter sentences when possible.

Response: Thank you for your constructive suggestions. In the revised manuscript, we have revised the longer sentences into shorter ones as recommended.

Add p values for Fig 2d and e.

Response: Thank you for your constructive suggestions. We have included the results of significance tests, adding *P*-values to the comparisons of isoform number across different methods in the revised manuscript.

Reviewer #2

Major concerns:

(1) As is imperative for all newly developed methods, cross comparisons are essential to make the new method valuable because developers always claim their tool is the most superior method. In the introduction and discussion sections, the authors repeatedly emphasize the deficiencies of short reads methods and third generation sequencing; it would help if the authors would provide more details and references for the deficiencies of the methods referenced. For example, third generation sequencing is expensive and it is unclear how much data needs to be generated and at what sensitivity level for this vs other algorithms to succeed comparatively.

Response: Thank you for the constructive suggestions.

First, in the revised manuscript, we cited the public viewpoint about the limitations of long-read based methods: "Long-read sequencing-based single-cell RNA sequencing methods can reveal full-length RNA isoforms, but the high cost, lower sequencing depth, and reduced sensitivity make them less applicable for integration into single-cell multi-omics analysis [4,5]" (**Line 45-48, Page 2**).

More importantly, during the revision of this manuscript, we obtained extra sequencing data by applying different methods in the same kind of cell types. We compared the performance of scRCAT-seq2 to other methods capable of detecting isoforms at the single-cell level, including PacBio long-read sequencing-based ScISOr-seq and HIT-scISO-seq, Nanopore long-read sequencing-based SCAN-seq2 and scCOLOR-seq, and short-read sequencing-based scRCAT-seq. We observed the highest sensitivity and efficiency to detect single-cell isoforms by scRCAT-seq2 across different sequencing depths of equal cost. The results show that the saturation sequencing depth for scRCAT-seq2 is 4 million reads (equal to 3.2 dollar), while for the Nanopore-based SCAN-seq it is 0.4 million (equal to 7), and for the PacBio sequencing-based scISOr-seq it is 0.4 million (equal to \$120) (**Fig. 2g, Supplementary Fig. 6d**). This indicates that scRCAT-seq2 has a significantly lower cost but higher read coverage.

In addition, we compared the number of isoforms detected by each method at the saturated sequencing depth. For low-throughput samples (ES cells), sequenced at a saturated depth, scRCAT-seq2 can detect $12,695 \pm 348$ isoforms, which is significantly more compared to the other approaches, including ScISOr-seq ($1,041 \pm 41$), SCAN-seq2 ($8,566 \pm 123$), and scRCAT-seq ($2,152 \pm 130$) (**Fig. 2h, Supplementary Fig. 6b**). For the high-throughput single-cell samples, the cDNA was amplified on microfluidic platforms, and scRCAT-seq2 also shows a higher number of isoforms compared to HIT-scISO-seq and Nanopore-based methods in monkey CEC (1192 ± 4.65 vs. 888 ± 9.06) and human retinal organoids (1989 ± 9.48 vs. 1133 ± 5.45) respectively (**Fig. 2i, Supplementary Fig. 6c**). These results indicate that scRCAT-seq2 outperforms the existing methods in terms of sensitivity and cost-effectiveness.

Most importantly, we have implemented an integrated analysis of isoform and accessibility, conducting in-depth analysis of epigenetic associations of isoform-level differential expression, a capability not yet achieved by other methods.

In summary, our comparisons with existing single-cell isoform methods reveal that scRCAT-seq2 offers better sensitivity and efficiency. It allows simultaneous analysis of accessibility and isoform expression within the same cell, providing an unprecedented new method for elucidating the isoform differences and chromatin accessibility at the single-cell level.

(2) Several methods with same or similar functions are mentioned in the manuscript, including, ScISOr-Seq, ATAC-seq/ ATAC-seq2. The authors should also include more recent methods developed in the last two years, including but not limited to HIT-scISOseq (PMID: 37149708) and Scasa (PMID: 34864849).

Response: Thank you for your constructive suggestions. In the revised manuscript, we incorporated data comparing our method with those developed in the past two years, including additional scRCAT-seq2 and nanopore sequencing data, and conducted comparative analyses of different sequencing methods.

Initially, we compared the performance of scRCAT-seq2 and other methods that use the PacBio sequencing platform to obtain full-length transcriptomes. Our data showed that

scRCAT-seq2 consistently detected a higher number of isoforms than scISOr-seq ($12,695 \pm 348$ vs. $1,041 \pm 41$), regardless of sequencing costs in the same cells. Furthermore, scRCAT-seq2 required significantly lower sequencing saturation costs than scISOr-seq (**Fig. 2h, Supplementary Fig. 6d**). Similarly, on the 10X platform, scRCAT-seq2 detected more isoforms than HIT-scISO-seq (1192 ± 4.65 vs. 888 ± 9.06) (**Fig. 2i**).

We also compared scRCAT-seq2 with Nanopore-based methods, including SCAN-seq2. The results showed that scRCAT-seq2 detected more isoforms than SCAN-seq2 at different sequencing costs, and at saturation ($12,695 \pm 348$ vs. $8,566 \pm 123$), scRCAT-seq2 identified significantly more isoforms than SCAN-seq2 (**Fig. 2h, Supplementary Fig. 6d**). Lastly, we performed parallel sequencing of D45 retinal organoids using scRCAT-seq2 and a Nanopore-based method, demonstrating that scRCAT-seq2 detected more isoforms (1989 ± 9.48 vs. 1133 ± 5.45) (**Fig. 2i**). In addition, compared to another nanopore based method scCOLOR-seq which can detect less than 500 genes per cell, scRCAT-seq2 also showed a higher efficiency.

Scasa, proposed by Pan et al., is an analysis tool for isoform-level quantification using the 10X 3' single-cell transcriptomics technology. While the accuracy and efficiency of Scasa is still largely limited due to lack of important features away from 3' end. In contrast, scRCAT-seq2 identifies full-length isoform features, including TSS, TES, and exons for each mRNA molecule, allowing the extraction of distinct transcriptomic features across both high and low throughput, and detects more isoforms than Scasa.

In summary, scRCAT-seq2 offers better sensitivity and efficiency than other methods in recent years.

(3) The accuracies were calculated based on SIRV-set 3 kits, which contains 69 isoforms. It is unclear why the authors chose this kit for their accuracy evaluation (and why only this kit). Some explanation is needed as this is a small RNA isoform set and accuracy is one of the most critical conditions of the new method.

Response: Thank you for your constructive suggestions. In the revised manuscript, we included a detailed rationale for selecting SIRV-Set3 for our accuracy assessment as

follows:

Real cellular-derived mRNA is highly complex and lacks a ground-truth for assessing the sensitivity and accuracy. We spiked in the RNA standard set (SIRV-set 3) containing 92 ERCCs and 69 isoforms to serve as a ground-truth control.

Four SIRV sets (Set1-4) are available, each comprising 69 SIRV isoforms. Their differences are shown in the table below. Their differences are shown in the table below. Set1 and Set2 differ only in the concentrations of these isoforms. Set3 and Set4 includes 92 ERCC isoforms at varying concentrations, providing a more comprehensive benchmark for assessing accuracy of quantification compared to Set1 and Set2. Although Set4 contains 15 more SIRV isoforms than Set3, but the difference is minimal (161 vs. 176 isoforms), we think Set 3 should be OK to be used as a control. (**Fig. 2c**). It turns out that ScRCAT-seq2 can identify 61 SIRV transcripts, with a sensitivity of 88%, and the full-length features were revealed for all the 61 transcripts detected (**Fig. 2b, c**).

		SIRV-Set 1	SIRV-Set 2	SIRV-Set 3	SIRV-Set 4
Cat. No		025.03	050.0*	051.0*	141.0*
Module(s)	Isoforms	Isoform Mixes E0, E1, E2	Isoform Mix E0	Isoform Mix E0	Isoform Mix E0
	ERCC	X	X	ERCC Mix 1	ERCC Mix 1
	long SIRVs	X	X	X	long SIRVs
Property	Isoform detection, and quantification	✓	✓	✓	✓
	Dynamic range	partially	X	✓	✓
	Length > 2.5 kb	X	X	X	✓
	Pipeline Validation	✓	partially	partially	partially
	Sample Control	X	✓	✓	✓
Number of spike-in transcripts in each mix		69 (69 isoforms in each Mix)	69 (69 isoforms)	161 (69 isoforms, 92 ERCCs)	176 (69 isoforms, 92 ERCCs, 15 long SIRV)

Table 1. Details of SIRV Set1-4 (<https://www.lexogen.com/sirvs/sirv-sets/>).

(4) When comparing the sensitivity to ScISOr-seq, the authors claimed that scRCAT-seq2 was demonstrated to detect a higher number of genes and isoforms. Please explicitly state which set/references are used in the comparisons.

Response: Thank you for your comment. In the revised manuscript, to clearly demonstrate the comparison between scRCAT-seq2 and other methods, we detailed the cell type and tissue information used for single-cell sequencing in the updated figures and methods. Additionally, we refined our analytical methods and included comprehensive details on the genomes and genome annotations used in our analyses. For instance, in the updated methods section, we specify the reference genomes and annotations used: "Reads were mapped to the reference genomes (human: hg38 and GENCODE v44, mouse: mm39 and GENCODE vM33, monkey: GCA_012559485.3 and GCF_012559485.2(v102) from NCBI)" (**Line 514-516, Page 19**).

(5) ScISOr-seq seems the only method applied in the cross comparison, as suggested in the previous points, the authors should assess the accuracies, sensitivities, and the functional detections on their human retinal organoids RNA set if possible to compare to previously developed methods.

Response: Thank you for your constructive suggestions. In the revised manuscript, we added sequencing data with human retinal organoids and monkey limbal epithelial cells to compare scRCAT-seq2 with previously developed methods based on nanopore and PacBio long-read sequencing.

By comparing the average number of isoforms detected within human retinal organoids, we demonstrated that scRCAT-seq2 identifies more isoforms at the single-cell level than nanopore sequencing (1989 ± 9.48 vs. 1133 ± 5.45) (**Fig. 2i**). Additionally, we found that at identical sequencing costs, scRCAT-seq2 detects more isoforms at the population level. Similarly, we observed that scRCAT-seq2 detects more isoforms than HIT-scISO-seq (1192 ± 4.65 vs. 888 ± 9.06) (**Fig. 2i**) in the monkey limbal epithelial cells.

In summary, our comparative analysis of different sequencing methods demonstrates that scRCAT-seq2 has higher sensitivity and efficiency than other sequencing methods for detecting single-cell isoforms.

Minor concerns:

It appears some of the paragraphs were written by AI tools based on anti-AI writing scanning programs. For example, the first paragraph (line 216-221), fourth and fifth paragraphs (line 243-266) in the discussion section, and first paragraph of introduction section (line 45-54), have very high probabilities for AI writings. While I have nothing against such tools being used, the authors should modify the written language generated to avoid this being picked up.

Response: Thank you for bringing this to our attention. We appreciate you taking the time to carefully review our manuscript and providing this valuable feedback. We have thoroughly reviewed these sections and have made substantial revisions to the writing style and phrasing to present the work in a clear and authentic manner, without any unintended use of AI-generated text.

Reviewer #3

(1) The author mentioned that they used two different strategies to co-profile mRNA and chromatin accessibility, one of them is using a 10x platform. The authors should provide more details about these protocols both using schematic figures and discussing details in the Methods section.

Response: Thank you for your constructive suggestions. In our revised manuscript, we updated the schematic diagrams to clearly illustrate the strategies based on nuclear-cytoplasmic separation (**Supplementary Fig. 1**) and the 10x platform (**Fig. 1**). Additionally, we discussed the similarities and differences between these two strategies in detail in the Methods section.

The similarities are that both strategies use a circularization approach that connects the 5' and 3' ends, enabling sequencing of individual mRNA molecules' transcription start site (TSS), transcription end site (TES), and exons.

The main differences are: The high-throughput approach using the 10x platform allows simultaneous detection of chromatin accessibility and isoform expression within the same cell, while the low-throughput approach involves chromatin and isoform detection through nuclear-cytoplasmic separation. To accommodate the 10x platform's barcode sequences, we modified the adapter primers used compared to the nuclear-cytoplasmic separation approach.

In summary, the revised schematic diagrams and the detailed discussion of the similarities and differences between the two strategies provide a clear illustration of the methodological advancements in our study.

(2) The authors compared their methods with ScISOr-seq, can they also compare with Nanopore sequencing of single-cell transcriptomes (scCOLOR-seq, Nature Biotechnology, volume 39, pages1517–1520 (2021))?

Response: Thank you for the constructive suggestions. In the revised manuscript, we included a comparison of scRCAT-seq2 with Nanopore-based single-cell transcriptome sequencing methods, involving both low-throughput SCAN-seq2 and high-throughput Nanopore sequencing.

First, we compared the performance of scRCAT-seq2 and SCAN-seq2. The results indicate that scRCAT-seq2 detects more isoforms and genes than SCAN-seq2. Additionally, at equivalent sequencing costs, scRCAT-seq2 detects more isoforms ($12,695 \pm 348$ vs. $8,566 \pm 123$) and reaches sequencing saturation at a lower cost compared to SCAN-seq2 (**Fig. 2g-h, Supplementary Fig. 6b**).

Next, we conducted simultaneous Nanopore and scRCAT-seq2 sequencing on human retinal organoids. scRCAT-seq2 detected more isoforms at both the single-cell (1989 ± 9.48 vs. 1133 ± 5.45) and cell-type levels compared to the Nanopore data (**Fig. 2i**). Furthermore, scRCAT-seq2 showed higher efficiency in detecting genes per cell compared to another Nanopore-based method, scCOLOR-seq, which can detect less than 500 genes per cell.

In summary, our comparative analysis demonstrates that scRCAT-seq2 has superior performance in detecting isoforms at the single-cell level compared to other Nanopore-based transcriptome sequencing methods.

(3) How robust is scRICA-seq? The authors should provide results by comparing technical replicates.

Response: Thank you for your constructive suggestions. In the revised manuscript, we assessed the robustness of the scRCAT-seq2 method.

First, we evaluated the technical reproducibility of scRCAT-seq2 using ERCC spike-in RNA. By analyzing the correlation between the expression levels of ERCCs across different sequencing libraries and their annotated concentrations, we found that all scRCAT-seq2 libraries had a correlation coefficient exceeding 0.95 (**Fig. 2d**).

Next, we analyzed the correlation of ERCC expression levels between different sequencing libraries, finding correlation coefficients consistently above 0.95 (**Fig. 2e**). Finally, we performed correlation analysis on scRCAT-seq2 data from different embryonic stem (ES) and 293T cells, and observed high consistency in gene expression (Pearson correlation coefficient = 0.89) (**Supplementary Fig. 5**).

Notably, compared to the SCAN-seq2 method, scRCAT-seq2 exhibited higher consistency for the single-cell sequencing data.

In summary, our results demonstrate that the scRCAT-seq2 method exhibits good technical reproducibility and robustness, with high consistency in gene expression profiles across different sequencing libraries and cell types.

(4) What is the throughput of scRICA-seq? How many cells it can process in each experiment? How many cells did they process in Figure 3-5?

Response: Thank you for your comment. We developed two distinct strategies to cater to different experimental needs.

The first strategy is for low-throughput samples, which can process between 1-96 cells per experiment. This approach is suitable for applications that require in-depth analysis of a small number of cells.

The second strategy is for high-throughput detection based on microfluidics platforms. In our tests, this approach was applied to D45 retinal organoids and monkey corneal margin epithelium using the 10X platform, identifying 12,344 and 8,238 cells, respectively.

In Figures 3-5, we analyzed three-dimensional information from a total of 12,344 cells. In the revised manuscript, we included detailed information on the number of cells processed in each experiment (**Supplementary Table 2**).

The two strategies provide flexibility to accommodate different experimental needs, from low-throughput in-depth analysis to high-throughput screening of large cell populations. This allows researchers to choose the most appropriate approach based on their specific research objectives.

(5) The authors developed scRCAT-seq2 using an optimized protocol of scRCAT-seq, and they should provide data showing how much improvement scRCAT-seq2 achieves.

Response: Thank you for your constructive suggestions. In the revised manuscript, we added data to compare the performance of the newly developed scRCAT-seq2 method with the earlier scRCAT-seq approach.

Firstly, we compared the gene body coverage between the two methods. The previously published scRCAT-seq method primarily targets sequencing of the transcription start

site (TSS) and transcription end site (TES), whereas scRCAT-seq2 can capture full-length features including TSS, TES, and exons in the middle of the isoforms (**Fig. 2a**). Secondly, we found that scRCAT-seq2 significantly outperforms scRCAT-seq in terms of the number of isoforms detected ($12,695 \pm 348$ vs. $2,152 \pm 130$) (**Fig. 2g-i**). Overall, our newly developed scRCAT-seq2 method significantly enhances our ability to detect isoforms by sequencing the TSS, TES, and exons of the same mRNA molecule. This improvement in isoform detection will enable more comprehensive and accurate transcriptome analysis at the single-cell level.

Fig. 2. Gene body coverage of scRCAT-seq.

(6) Figures 3 and 4 are very confusing. What protocols did the authors use for scATAC-seq + scRNA-seq? What is the purpose of Figures 3 and 4?

Response: Thank you for the opportunity to clarify the revisions made to Figures 3 and 4. We have made substantial updates to these figures to better showcase the single-cell multiomics landscape of human retinal organoids and the dynamic changes and correlations between gene expression (scRNA-seq), isoforms (scRCAT-seq2), and chromatin accessibility (scATAC-seq) during retinal cell lineage development.

Specifically, Figure 3 now compares the cell clustering results obtained from scRCAT-seq2 to the 10x Multiomics sequencing data, which includes both scRNA-seq and scATAC-seq. This demonstrates the high consistency between scRCAT-seq2 and the

10x Multiomics approach in cell classification, and highlights the cell-type-specific molecular features across these three dimensions.

Fig. 4 has been further expanded to analyze the correlations between changes in chromatin accessibility and differential isoform expression during the differentiation of retinal progenitor cells (RPCs) towards the photoreceptor lineage. This reveals the potential regulatory role of chromatin accessibility on isoform-level regulation of fate-determining factors, and describes the relationships between the chromatin accessibility of these fate-determining factor binding sites and the expression of their downstream target genes.

In summary, these revised figures and the accompanying text demonstrate how the novel scRCAT-seq2 method can be used for in-depth exploration of the dynamic changes and interrelationships of gene expression, isoform regulation, and chromatin accessibility during retinal cell differentiation processes. We hope this clarifies the substantial updates made to these key figures.

(7) Line 204-209, the authors did not provide the chromatin accessibility data in Extended Data Fig. 6e-j, how did they conclude that “we found that the majority of splicing changes were independent of chromatin accessibility”?

Response: Thank you for your comment. In the revised manuscript, our analysis revealed that 7.43% of the splicing sites exhibited significant changes in chromatin accessibility during retinal progenitor cell differentiation into cone photoreceptors (**Fig. 5b**). By performing transcription factor motif analysis, we predicted the regions with increased chromatin accessibility near the dynamic splicing sites were enriched with 78 transcription factors, while the regions with decreased chromatin accessibility were enriched with 57 transcription factors (**Supplementary Fig. 16**). Among the predicted transcription factors, we identified 11 that have been previously reported to be directly involved in regulating RNA splicing, including NEUROD1, SP1, and KLF4.

(8) I don't quite understand Figure 5k and l, how to read these figures?

Response: Thanks for the comments. To more clearly demonstrate the relationship

between isoform alternative splicing and chromatin accessibility, we have modified the organization of the figures to show the association between alternative splicing (AS) events, chromatin accessibility, and transcription factor (TF) binding in representative examples.

For instance, the MTHFD2 gene has an isoform1 (ENST00000477455) that is mainly expressed in retinal progenitor cells (RPCs), while isoform2 (ENST00000394053) is predominantly expressed in cone cells. There is an alternative splicing event in exon 3 between these two isoforms, and this exon region shows higher chromatin accessibility in cones compared to RPCs, indicating the association of alternative splicing with differential chromatin accessibility between RPCs and cones (**Fig. 5e**). Furthermore, we predict the binding of splicing-related transcription factors ATF2, KLF4, and NEUROD1 at this site, and these TFs are observed to be differentially expressed between RPCs and cones. This suggests that both the expression levels of the transcription factors and the chromatin accessibility of their binding sites jointly regulate the differential splicing of MTHFD2 (**Fig. 5f**).

We hope this revised description helps the reader better understand the association between chromatin accessibility, the expression of alternative splicing-related genes, and the regulation of RNA splicing during retinal cell differentiation.

(9) The data analysis in Figure 5 is too vague. For example, Line 212, "This finding suggests a potential relationship between chromatin accessibility and RNA splicing."

Response: Thank you for your comments. Thank you for your comments. As described earlier, we have made extensive modifications and improvements to **Fig. 5** to present the results more clearly. Through our analysis, we found that both differential chromatin accessibility and the differential expression of AS-related transcription factors, including NEUROD1, KLF4, ATF2, and others, are associated with the RNA splicing differences between RPC and Cones. We hypothesize that chromatin accessibility and the expression of AS-related transcription factors may jointly influence the recruitment of splicing factors, thereby regulating mechanisms underlying photoreceptor development and fate determination.

(10) The detailed code for data analysis is missing.

Response: Thank you for pointing out this issue. We have uploaded the detailed code for data analysis to our GitHub repository (<https://github.com/huyoujinlab/scRCAT-seq2>). We have also included a statement in the "CODE AVAILABILITY" section of our manuscript detailing the availability of this code.

Reviewers' Comments:

Reviewer #1:

Remarks to the Author:

My comments are fully addressed by the revised manuscript and I would like to thank the authors for this effort.

Reviewer #2:

Remarks to the Author:

The authors have been responsive to comments and critique raised and have updated and improved the manuscript significantly. I have no further concerns at this point.

Reviewer #3:

Remarks to the Author:

The authors have addressed all my concerns, and I would recommend the paper be accepted for publication.

Comments from the editor

(1) First, we ask you to revise your paper to address our editorial requests (in the attached Author Checklist) and any remaining comments from reviewers (included at the end of this email, if applicable). In addition, please cite a previously published paper (PNAS 2012: <https://www.pnas.org/doi/10.1073/pnas.1217322109>), which reported a method called PMA (and another method SMA), and have proper discussions of the previous methods.

Response: Thank you very much for the constructive suggestions.

We cited PNAS 2012: <https://www.pnas.org/doi/10.1073/pnas.1217322109>, and provided an appropriate discussion on PMA and SMA, both of which are sequencing methods for full-length cDNA. Consequently, we have expanded and revised them within the discussion. While a few methods can capture and amplify the full-length cDNA of single cells, including smart-seq, smart-seq2, smart-seq3, PMA and SMA, but due to the incapability to link internal short reads to the UMI located in the 5'/3' end, the efficiency to identify full-length isoforms is still largely limited. (**Discussion, Pan et al., 2012**)

Comments from the reviewers

Reviewer #1:

My comments are fully addressed by the revised manuscript and I would like to thank the authors for this effort.

Response: Thank you very much for your kind feedback. We are glad to hear that our revisions have addressed your comments. Your support and insights have been invaluable in improving our manuscript.

Reviewer #2:

The authors have been responsive to comments and critique raised and have updated and improved the manuscript significantly. I have no further concerns at this point.

Response: We deeply appreciate your positive feedback and acknowledgment of our efforts to address the comments and critiques raised. Your guidance and feedback have been instrumental in helping us refine our work, and we are grateful for your support.

Reviewer #3:

The authors have addressed all my concerns, and I would recommend the paper be accepted for publication.

Response: Thank you once again for your time and expertise.